# Offline-to-online Reinforcement Learning for Image-based Grasping with Scarce Demonstrations

## Abstract

Offline-to-online reinforcement learning (O2O RL) aims to obtain a continually improving policy as it interacts with the environment, while ensuring the initial policy behaviour is satisficing. This satisficing behaviour is necessary for robotic manipulation where random exploration can be costly due to catastrophic failures and time. O2O RL is especially compelling when we can only obtain a scarce amount of (potentially suboptimal) demonstrations—a scenario where behavioural cloning (BC) is known to suffer from distribution shift. Previous works have outlined the challenges in applying O2O RL algorithms under the image-based environments. In this work, we propose a novel O2O RL algorithm that can learn in a real-life image-based robotic vacuum grasping task with a small number of demonstrations where BC fails majority of the time. The proposed algorithm replaces the target network in off-policy actor-critic algorithms with a regularization technique inspired by neural tangent kernel. We demonstrate that the proposed algorithm can reach above 90% success rate in under two hours of interaction time, with only 50 human demonstrations, while BC and existing commonly-used RL algorithms fail to achieve similar performance.

## 1 Introduction

Imitation learning (IL) is a popular method for robot learning partly due to the wider data availability, improved data collection techniques, and the development of vision language models (Padalkar et al., 2023; Zhao et al.; Haldar et al., 2024). However, while these approaches are more robust to various manipulation tasks, the training requires abundant demonstration data. For niche robotic applications where data is scarce, supervised IL methods such as behavioural cloning (BC) are known to suffer from distribution shift (Ross et al., 2011; Rajaraman et al., 2020) and more generally cannot perform better than the demonstrator (Xu et al., 2020; Ren et al., 2021). Alternatively, we focus on offline-to-online reinforcement learning (O2O RL), which is a two-step algorithm that first pretrains a policy followed by continual improvement with online interactions (Song et al., 2023; Nakamoto et al., 2023; Tan & Xu, 2024).

The first step, known as offline RL (Levine et al., 2020), has made tremendous progresses on state-based environments (Kumar et al., 2020; Yu et al., 2020; Fujimoto & Gu, 2021; Tangri et al., 2024), but it has recently been observed that the transfer of algorithms to image-based environments can be challenging (Lu et al., 2023; Rafailov et al., 2024) (we also provide an example in Appendix B). Naturally, some have investigated the potential benefits of pretrained vision backbones, pretraining self-supervised objectives, and data-augmentation techniques (Hansen et al., 2023; Li et al., 2022; Hu et al., 2023). These directions are also investigated in the online RL setting (Sutton, 2018) concurrently, which has seen more successes with image-based domains in both simulated and real-life environments (Singh et al., 2020; Yarats et al., 2022; Wang et al., 2022; Luo et al., 2024). Nevertheless, these algorithms still leverage large amount of data and may require a long-duration of data collection in real life. Our question is **whether we can stabilize visual-based RL algorithms to enable sample-efficient RL on real-life robotics task**.

As far as we know there has been very limited success in applying RL on real-life image-based robotic manipulation without using any simulation (Luo et al., 2024; Seo et al., 2024; Hertweck

et al., 2020; Lampe et al., 2024). One potential reason for this limitation is due to its instability in the learning dynamics. Specifically, most empirically sample-efficient algorithms are off-policy actor-critic based that involve learning a Q-function (Fujimoto et al., 2018; Haarnoja et al., 2018; Hiraoka et al., 2022a; Chen et al., 2021; Ji et al., 2024), which has been observed to be unstable— the Q-values tend to diverge (Baird, 1995; Yang et al., 2022) and overestimate (Hasselt, 2010). Recent work has shown that in deep RL, Q-divergence is correlated to the similarity of the latent representation between state-action pairs (Kumar et al., 2022; Yue et al., 2023). Particularly the latent representation learned by the neural networks has a high correlation between in-distribution and out-of-distribution transitions, resulting in Q-value divergence (Kumar et al., 2022). Yue et al. (2023) has also provided theoretical analyses to explain this phenomenon through the lens of neural tangent kernel (Jacot et al., 2018). We claim that **addressing this Q-divergence is a critical step to addressing our question**.

To this end, we develop an O2O RL algorithm that enables training policies on a real-life image-based robotic task in a short amount of time, under two hours, with only limited offline demonstrations. In this scenario the amount of offline demonstrations is insufficient for training a good behaviourally-cloned policy. We make the following contributions: (1) We propose a method that simplifies Q-learning by replacing the target network with a regularization term that is inspired by neural tangent kernel (NTK). We refer our method as Simplified Q. (2) We conduct experiments on three simulated manipulation environments and a real-life image-based grasping task, and compare Simplified Q against behavioural cloning and multiple existing RL algorithms. In this case the compared algorithms are unable to achieve similar performance on the real-life environment even with same amount of total data. (3) We show that vision backbone pretraining is unnecessary for performant offline-to-online transfer. (4) We provide ablation studies to demonstrate the importance of the NTK regularizer.

## 2 PRELIMINARIES AND BACKGROUND

### 2.1 PROBLEM FORMULATION

**Markov Decision Process.** A reinforcement learning (RL) problem can be formulated as an infinite-horizon Markov decision process (MDP) $\mathcal{M} = (\mathcal{S}, \mathcal{A}, r, P, \rho_0, \gamma)$, where $\mathcal{S}$ and $\mathcal{A}$ are respectively the state and action spaces, $r : \mathcal{S} \times \mathcal{A} \to [0, 1]$ is the reward function, $P \in \Delta_{\mathcal{S} \times \mathcal{A}}^{\mathcal{S}}$ is the transition distribution, $\rho_0 \in \Delta^{\mathcal{S}}$ is the initial state distribution, and $\gamma \in [0, 1)$ is the discount factor. A policy $\pi \in \Delta_{\mathcal{S}}^{\mathcal{A}}$ can interact with the MDP $\mathcal{M}$ through taking actions, yielding an infinite-length random trajectory $\tau = (s_0, a_0, r_0, s_1, a_1, r_1, \cdots)$, where $s_0 \sim \rho_0, a_t \sim \pi(s_t), s_{t+1} \sim P(s_t, a_t), r_t = r(s_0, a_0)$. The return for each trajectory is $G = \sum_{t=0}^{\infty} \gamma^t r_t \leq \frac{1}{1-\gamma}$. We further define the value function and Q-function respectively to be $V_\gamma^\pi(s) := \mathbb{E}_\pi [\sum_{t=0}^{\infty} \gamma^t r_t | s_0 = s]$ and $Q_\gamma^\pi(s, a) := \mathbb{E}_\pi [\sum_{t=0}^{\infty} \gamma^t r_t | s_0 = s, a_0 = a]$. The goal is to find a policy $\pi^*$ that maximizes the expected cumulative sum of discounted rewards for all states $s \in \mathcal{S}$, i.e. $\pi^*(s) = \arg\max_\pi V_\gamma^\pi(s)$.

**Offline-to-online Reinforcement Learning.** RL algorithms require exploration which is often prohibitively long and expensive for robotic manipulation. To this end, we consider offline-to-online (O2O) RL, a setting where we are given an offline dataset that is generated by a potentially suboptimal policy. Generally, we can decompose O2O RL into two phases: (1) pretraining an offline agent using offline data, and (2) continually training the resulting agent through online interactions. Our goal is to leverage this offline dataset and a limited number of online interactions to train an agent that can successfully complete the task. We consider the setting where we also include offline data during the online interaction (Tan & Xu, 2024; Huang et al., 2024).

In this work, we assume access to $N$ trajectories truncated at $T$ timesteps $\mathcal{D}_{\text{off}} = \{(s_0^{(m)}, a_0^{(m)}, r_0^{(m)}, \ldots, s_{T-1}^{(m)}, a_{T-1}^{(m)}, r_{T-1}^{(m)}, s_T^{(m)})\}_{m=1}^M$ as the offline dataset, and denote $\mathcal{D}_{\text{on}}$ as the interaction buffer for data collected during the online phase. We refer $\mathcal{D}$ as the total buffer that samples from both $\mathcal{D}_{\text{off}}$ and $\mathcal{D}_{\text{on}}$ with equal probability, a technique known as symmetric sampling (Ball et al., 2023). We highlight that offline datasets with truncated trajectories are natural in the robotics setting as real-systems are setup for human demonstrators to gather one trajectory at a time to minimize switching between controllers.

## 2.2 CONSERVATIVE Q-LEARNING

We build our algorithm on conservative Q-learning (CQL) (Kumar et al., 2020). CQL imposes a pessimistic Q-value regularizer on out-of-distribution (OOD) actions to mitigate unrealistically high-values on unseen data. Suppose the Q-funciton $Q_\theta$ is parameterized by $\theta$, the CQL training objective is defined by:

$$\mathcal{L}_{\text{CQL}}(\theta) := \alpha \left( \mathbb{E}_{\mathcal{D},\mu} \left[ Q_\theta(s,a') \right] - \mathbb{E}_{\mathcal{D}} \left[ Q_\theta(s,a) \right] \right) + \frac{1}{2} \mathbb{E}_{\mathcal{D}} \left[ \left( Q_\theta(s,a) - \mathcal{B}_N^\pi \bar{Q}(s,a) \right)^2 \right], \quad (1)$$

where $\alpha$ is a hyperparameter controlling strength of the pessimistic regularizer, $\mu$ is an arbirary policy, $a' \sim \mu(s)$, $a \sim \mathcal{D}$, and $\mathcal{B}_N^\pi \bar{Q}(s,a) := \sum_{n=0}^{N-1} \gamma^n R_n(s,a) + \gamma^N \mathbb{E}_\pi \left[ \bar{Q}(s',a') \right]$ is the $N$-step empirical Bellman backup operator applied to a delayed target Q-function $\bar{Q}$, writing $\mathcal{B}_1^\pi = \mathcal{B}^\pi$ for conciseness. When $\bar{Q} = Q$, we use semi-gradient which prevents gradient from flowing through the objective. The first term is the pessimistic Q-value regularization and the second term is the standard $N$-step temporal-difference (TD) learning objective (Sutton, 2018). There can be multiple implementations of CQL. Common implementation builds on top of soft actor-critic (Singh et al., 2020; Haarnoja et al., 2018). Alternatively, crossQ (Bhatt et al., 2024) has demonstrated that we can replace the delayed target Q-function with the current Q-function by properly leveraging batch normalization (Ioffe, 2015). Calibrated Q-learning (Cal-QL) (Nakamoto et al., 2023) further augments the regularizer to only penalize OOD actions when their corresponding Q-values exceed the value induced by the dataset $\mathcal{D}$.

## 3 STABILIZING Q-LEARNING VIA DECOUPLING LATENT REPRESENTATIONS

It is desirable for RL algorithms to be stable and sample-efficient in robotic manipulation tasks—we propose an algorithm that encourages both properties. To address the former, we leverage ideas from the neural tangent kernel (NTK) literature and propose a regularizer to decouple representation during Q-learning. For the latter we leverage symmetric sampling from reinforcement learning with pretrained data (RLPD) (Ball et al., 2023) to encourage the agent to learn from positive examples. We build our algorithm on conservative Q-learning (CQL) (Kumar et al., 2020) to enable offline training and further include our proposed regularizer described in this section.

Q-learning algorithms are known to diverge (Baird, 1995) and suffer from the overestimation problem (Hasselt, 2010) even with double-Q learning (Yue et al., 2023; Ablett et al., 2024). Recent work leverages NTK to analyze the learning dynamics of Q-learning (Yue et al., 2023; Kumar et al., 2022; Yang et al., 2022; He et al., 2024; Ma et al., 2023; Tang & Berseth, 2024)—the Q-function of a state-action pair $(s',a') \in \mathcal{S} \times \mathcal{A}$ can be influenced by the Q-learning update of another state-action pair $(s,a) \in \mathcal{S} \times \mathcal{A}$. Let $\theta$ and $\theta'$ be the parameters of the Q-function before and after a stochastic gradient descent (SGD) step respectively, and define $\kappa_\theta(s,a,s',a') = \nabla_\theta Q_\theta(s',a')^\top \nabla_\theta Q_\theta(s,a)$. By performing SGD on the TD learning objective (second term in equation 1) with state-action pair $(s,a)$, we can write the Q-value of another state-action pair $(s',a')$ after the gradient update as

$$Q_{\theta'}(s',a') = Q_\theta(s',a') + \kappa_\theta(s,a,s',a') \left( Q_\theta(s,a) - \mathcal{B}^\pi \bar{Q}(s,a) \right) + \mathcal{O}(\|\theta'-\theta\|^2). \quad (2)$$

Here, $\kappa_\theta$ is known as the neural tangent kernel (Jacot et al., 2018) and the last term approaches to zero as the dimensionality approaches to infinity, a phenomenon known as lazy training (Chizat et al., 2019). Intuitively, a small magnitude in $\kappa_\theta(s,a,s',a')$ will result in $Q_{\theta'}(s',a')$ being less influenced by the update induced by $(s,a)$.

Suppose now the Q-function is parameterized as a neural network $Q_\theta(s,a) := w^\top \Phi(s,a)$ (i.e. last layer is a linear layer), where $\theta = [w, \Phi]$, $w$ is the parameters of the last layer, and $\Phi(s,a)$ is the output of the second-last layer, we can view $\Phi(s,a)$ as a representation layer. Thus, freezing the representation layer during Q-learning update, we can write equation 2 as

$$Q_{w'}(s',a') = Q_w(s',a') + \Phi(s',a')^\top \Phi(s,a) \left( Q_w(s,a) - \mathcal{B}^\pi \bar{Q}(s,a) \right) + \mathcal{O}(\|w'-w\|^2). \quad (3)$$

Kumar et al. (2022) is among the first to propose regularizing the representation layers of the Q-function with $R(\Phi) = \mathbb{E}_{\mathcal{D}} \left[ \Phi(s,a)^\top \Phi(s',a') \right]$, where $(s,a,s',a') \sim \mathcal{D}$ is the current and next state-action pairs from the buffer. Follow-up works modify the network architecture to include various normalization layers (Yang et al., 2022; Yue et al., 2023) and different regularizers (He et al., 2024; Ma et al., 2023; Tang & Berseth, 2024).

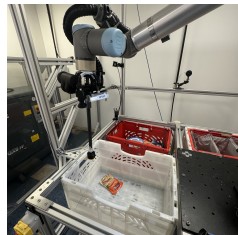 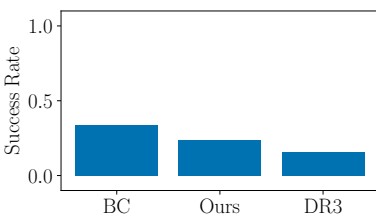 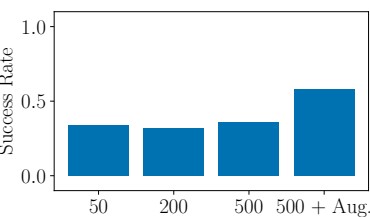

Figure 1: **(Left)** Image-based grasping environment setup. The agent is required to control the UR10e arm with vacuum suction to grasp the orange rice bag inside the bin and lift it well above the bin. **(Middle)** Comparison between BC and offline RL trained with Simplified Q (Ours). Simplified Q is able to grasp with limited success while BC performs marginally better than Simplified Q. **(Right)** The impact of offline dataset size on BC. Here BC is only able to achieve around 35% success rate until we further include image augmentation from Yarats et al. (2021). The success rates of various offline-trained policies. Each policy is evaluated on 50 grasp attempts.

Alternative approaches to mitigate this Q-divergence include using target network (Mnih et al., 2013) and double Q-learning (Hasselt, 2010). The former is introduced to reduce the influence of rapid changes of nearby state-action pairs during bootstrapping, which can also be addressed by decorrelating their corresponding features. Consequently, we remove the target network and introduce a regularizer that aims to decouple the representations between different state-action pairs. Specifically, our regularizer is defined to be

$$\mathcal{L}_{\mathrm{reg}}(\Phi) := \mathbb{E}_{s \sim \mathcal{D}, s' \sim \mathcal{D}} \left[ \left( \Phi(s, a_u)^\top \Phi(s', a'_\pi) \right)^2 \right], \tag{4}$$

where $s, s'$ are states sampled independently from the buffer $\mathcal{D}$, $\Phi(s, a)$ is the latent representation from the second-last layer of the model, $a_u \sim \mathcal{U}(\mathcal{A})$ is a uniformly sampled action, and $a'_\pi \sim \pi(s')$ is the next action sampled following the current policy. In other words, we are approximately orthogonalizing the latent representations through minimizing the magnitude of their dot products. Intuitively, equation 4 decorrelates the representations of any two states regardless of the current action, thereby minimizing the influence on other Q-values due to the current update. We note that in previous work, Kumar et al. (2022) has suggested a regularizer that takes the dot product between the latent representations which can cause the NTK to be negative—a Q-function update of a particular state-action pair may unlearn the Q-values of another state-action pair. Secondly our regularizer applies on not only the consecutive state-action pairs which decouples more diverse state-action pairs. Finally, the complete objective for updating the Q-function is

$$\mathcal{L}_Q(\theta) := \mathcal{L}_{\mathrm{CQL}}(\theta) + \beta \mathcal{L}_{\mathrm{reg}}(\Phi),$$

where $\beta > 0$ is a coefficient that controls the strength of the decorrelation. We call our method *Simplified Q* as we simplify Q-learning by leveraging this insight to remove existing components, as opposed to other methods where they further introduce extra components to improve learning (Yue et al., 2023; Tang & Berseth, 2024). Replacing the target network with this regularizer reduces the number of model parameters by half, and further reduces the number of hyperparameters to be a single scalar $\beta$, as opposed to having to tune the update frequency and the polyak-averaging term.

## 4 EXPERIMENTS

We aim to answer the following questions with empirical experiments: **(RQ1)** Can we perform offline RL using our proposed method? How does it compare with behavioural cloning (BC)? **(RQ2)** Can our proposed method enable O2O RL on real-life robotic manipulation? What about existing commonly-used RL algorithms? **(RQ3)** Can O2O RL eventually exceed imitation learning approaches with similar amount of data? **(RQ4)** How important is it to pretrain a vision backbone? **(RQ5)** Can the proposed NTK regularizer alleviate Q-divergence? We provide extra ablation results and our method's zero-shot generalization capability in Appendix C.

**Environment Setup.** We conduct our experiments on a real-life image-based grasping task (Figure 1). The task consists of controlling a UR10e arm to grasp an item inside a bin. The agent

Table 1: The average success rate of offline learning algorithms in three robomimic environments (Mandlekar et al., 2022). We evaluate each algorithm on low dimensional proficient-human (PH) and low dimensional multi-human (MH) datasets. As done in (Mandlekar et al., 2022) we run each algorithm on three seeds and report the best performance per seed. Bolded text means highest mean. Generally Simplified Q performs better than CQL except for one task. Having a wider critic for Simplified Q appears to stabilize learning and can further improve performance on some tasks.

|  | BC | CQL | Simplified Q (Ours) | Ours w/ Wider Critic |
|---|---|---|---|---|
| Lift (MH) | $\mathbf{100.00 \pm 0.00}$ | $82.00 \pm 6.18$ | $98.00 \pm 1.63$ | $99.33 \pm 0.54$ |
| Can (MH) | $\mathbf{84.00 \pm 2.49}$ | $26.67 \pm 5.68$ | $35.33 \pm 14.62$ | $37.33 \pm 4.65$ |
| Square (MH) | $\mathbf{47.33 \pm 0.54}$ | $1.33 \pm 1.09$ | $5.33 \pm 2.18$ | $12.00 \pm 2.49$ |
| Lift (PH) | $\mathbf{100.00 \pm 0.00}$ | $95.33 \pm 3.81$ | $78.67 \pm 16.61$ | $100.00 \pm 0.00$ |
| Can (PH) | $\mathbf{94.67 \pm 1.09}$ | $37.33 \pm 4.84$ | $87.33 \pm 3.57$ | $91.33 \pm 2.88$ |
| Square (PH) | $\mathbf{82.00 \pm 0.94}$ | $4.67 \pm 0.54$ | $7.33 \pm 3.03$ | $6.67 \pm 5.44$ |

observes a $64 \times 64$ RGB image with an egocentric view, the proprioceptive information including the pose and the speed, and the vacuum pressure reading. The agent controls the arm at 10Hz frequency through Cartesian velocity control with vacuum action—a 7-dimensional action space. The agent can attempt a grasp for six seconds. The agent receives a $+1$ reward upon grasping the item and moving it above a certain height threshold and a $+0$ reward otherwise. In the former, there is a randomized-drop mechanism to randomize the item location, otherwise a human intervenes and changes the item location. The attempts are fixed at six seconds (i.e. episode does not terminate upon success) unless the arm has experienced a protective stop (P-stop)—this enforces the agent to also learn to continually hold the item upon grasping it. This environment has a challenging exploration problem as we impose minimal boundaries—random policies can easily deviate away from the bin and go further above and outside the bin. Consequently in this O2O RL setting the policy must learn to leverage the offline data to accelerate learning.

**Baselines.** We compare Simplified Q against behavioural cloning (**BC**), and conservative Q-learning built on soft actor-critic (**SAC**) (Haarnoja et al., 2018) and crossQ (**CrossQ**) (Bhatt et al., 2024). SAC uses a target Q-network to stabilize learning while CrossQ removes the target Q-network by including batch normalization (Ioffe, 2015) in the Q-networks. To verify other existing Q-stabilization techniques we also include DR3 (**DR3**) (Kumar et al., 2022) and SAC with LayerNorm (**LN**) (Yue et al., 2023). The former uses a similar NTK regularizer as ours but only considers consecutive state-action pairs and without the squaring the dot product, while the latter uses LayerNorm to enable a better-behaved Q-function. For offline training, we provide 50 successful single-human-teleoperated demonstrations and train each policy for 100K gradient steps. During the online learning phase, we further run the RL algorithms for 200 episodes which corresponds to less than two hours of the total running time, combining both the interaction time and the learning update time. We note that the interaction time takes up only 20 minutes while the remaining time is for performing learning updates. For all RL algorithms we enable 3-step Q-learning and symmetric sampling for fairness, and run on three random seeds. All RL algorithms use a frozen image encoder that is pretrained trained with first-occupancy successor representation under Hilbert space (Moskovitz et al., 2022; Park et al., 2024). We provide further algorithmic and implementation details on the real-life experiments in Appendix E.

## 4.1 MAIN RESULTS

We first address RQ1 by training a BC policy and offline RL policies. We start with conducting preliminary experiments on simulated robotic manipulation tasks with low-dimensional observations from robomimic (Mandlekar et al., 2022)[1]. We compare three algorithms, Simplified Q, BC, and CQL, on lift, can, and square environments with proficient-human (PH) and multi-human (MH) demonstrations. Here Simplified Q uses the exact same implementation as CQL but with the target network replaced with our NTK regularizer with $\beta = 0.1$. We further include Simplified Q with

---

[1]Our code is available here: `https://anonymous.4open.science/r/robomimic-3DB6/robomimic/`

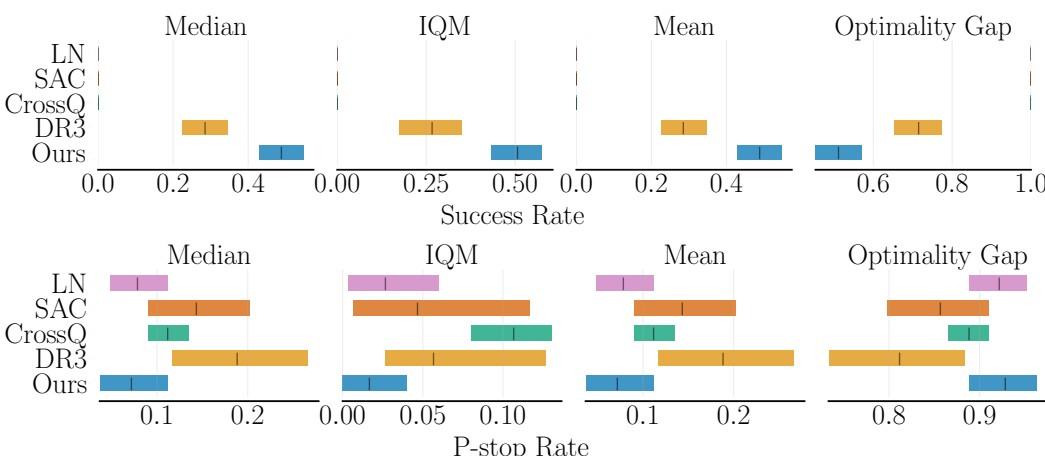

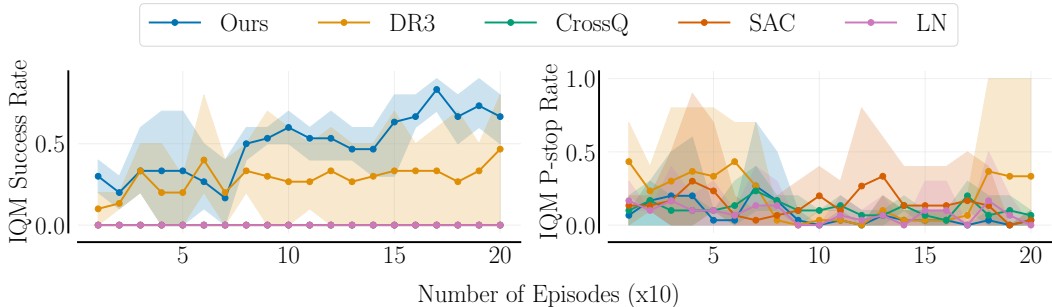

Figure 2: Aggregated success rate **(Top)** and P-stop rate **(Bottom)** across three seeds with 95% confidence intervals (CIs) (Agarwal et al., 2021). Simplified Q (Ours) performs better than DR3 in both success rates and P-stop rate generally. Furthermore, DR3 obtains significantly wider CIs compared to Simplified Q in both success rate and P-stop rate.

Figure 3: Success rate **(Left)** and P-stop rate **(Right)** across three seeds, averaged at every 10 episodes. Results are shown as an interquartile mean and shaded regions show 95% stratified boot-strap confidence intervals (CIs) (Agarwal et al., 2021). Simplified Q consistently achieves higher success rate and lower P-stop rate as amount of online interaction increases. While DR3 can achieve reasonable success rates, its CI is significantly wider than that of Simplified Q.

a wider critic where each layer consists of 2048 hidden units—Bhatt et al. (2024) has previously shown to improve performance. From Table 1 we observe that BC is superior than all other methods regardless of the task. Simplfied Q can outperform CQL on lift environment with the MH demon-strations and on can environment with the PH demonstrations, and can perform comparably on the remaining tasks. It appears that one run of Simplified Q in lift with PH data has diverged early in training which causes the wider standard error. However, Simplified Q with a wider critic appears to have smaller standard error compared to the default critic. This suggests that using the target network is not necessary for learning a good offline RL agent.

We now turn to our real-life robotic manipulation environment with image observations. Here all RL algorithms use a pretrained image encoder to accelerate training in wall time while BC is trained end-to-end. We then evaluate each policy on 50 grasp attempts. Figure 1, left, shows that BC agent can achieve 34% success rate. On the other hand, offline RL with Simplified Q and DR3 can achieve 24% and 16% success rates respectively. Although these three policies fail to pick most of the time, behaviourally the policies reach into the bin 100% of the time, demonstrating satisficing behaviours. On the other hand, offline RL policies that are trained using CrossQ, SAC, and LN achieve 0% success rate and totally fail to learn similar behaviours as the satisficing policies (we omit CrossQ, SAC, and LN results in Figure 1, left). In the offline setting we can see that Simplified Q can perform better than multiple offline RL algorithms but remains to be inferior compared to BC.

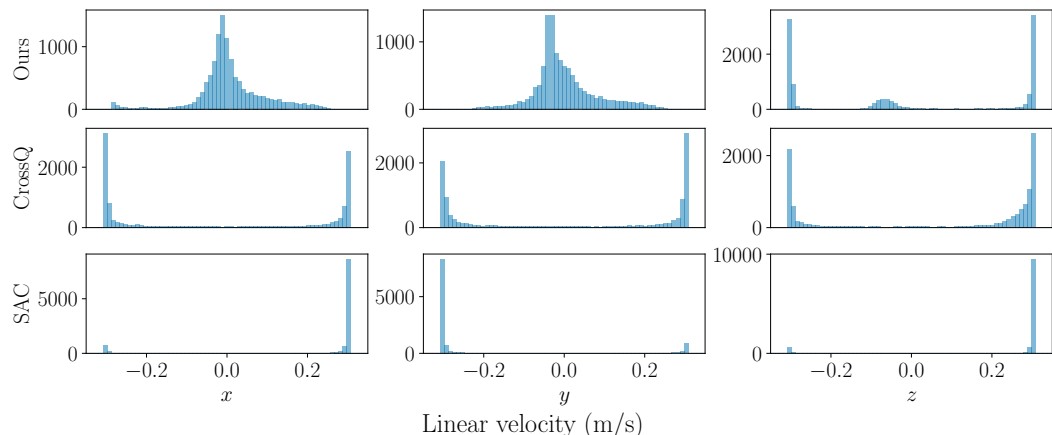

Figure 4: The frequency of actions being taken by the policy during training. We compare Simplified Q (Ours), CrossQ, and SAC. Our policy appears to be able to perform fine-grained actions on the $xy$ axes while CrossQ and SAC exhibits bang-bang behaviours. SAC further appears to have converged into moving towards a single direction.

To investigate the benefit of training with additional online interactions (RQ2), we deploy these offline-trained RL policies to the environment. CrossQ, SAC, and LN are unable to grasp the item at all, while Simplified Q and DR3 can immediately grasp the item. Notably, the latter two can also continually learn during the online phase. From Figure 2 we observe that Simplified Q is generally less susceptible to randomness and consistently achieve higher success rate over 200 online episodes when compared to DR3. Simplified Q is also safer in the sense that it experiences less P-stops compared to DR3. Inspecting the learning curves in Figure 3, we can clearly see that Simplified Q has a steeper increasing slope in the success rate with tighter confidence intervals than DR3. Furthermore the P-stop rate decreases over training as the policy becomes more adept. We speculate the main reason behind this is because DR3 considers only the state-action pairs from the current and next timesteps, while our approach considers any state-action pairs, thereby decorrelating more diverse state-action pairs.

Now, focusing on the behaviour of the policies of CrossQ, SAC, and Simplified Q, we observe that SAC performs the worst in particular as it has learned to go towards a specific corner in the workspace, away from the bin, which causes the arm to nearly reach singularity. CrossQ is able to hover around the bin but cannot reach and pick up the item successfully. Further visualizing the frequency of an action being taken by each policy during training in Figure 4, SAC has converged to move towards a particular direction and CrossQ has learned to perform bang-bang actions. On the other hand, Simplified Q has learned to perform fine-grained actions on linear $xy$ velocities, demonstrating more precise motion. We provide a visualization of the trajectories throughout training in Figure 8.

Finally, comparing Simplified Q with BC, it appears that the BC policy experiences distribution shift due to the lack of demonstrations. We observe in Figure 1, right, that it requires 500 demonstrations with image-augmentation techniques (Yarats et al., 2021) in order to achieve above $60\%$ success rate. On the other hand, Simplified Q can achieve near $70\%$ success rate within 200 episodes, suggesting O2O RL can indeed outperform BC without more data (RQ3).

## 4.2 THE IMPORTANCE OF PRETRAINED IMAGE ENCODER

One may argue that leveraging the pretrained image encoder might have enabled the sample efficiency of Simplfied Q (RQ4). To this end we also train an O2O RL agent end-to-end (E2E) using Simplfied Q. We also include a frozen randomly initialzed image encoder as a baseline. Figure 5, middle, demonstrates that the E2E RL agent can achieve similar performance as using a pretrained image encoder, furthermore both RL agents have learned to pick with limited success after the offline phase. On the other hand, while the agent using a frozen randomly-initialized encoder can pick the item up sometimes, it cannot further improve as it gathered more transitions. This suggests

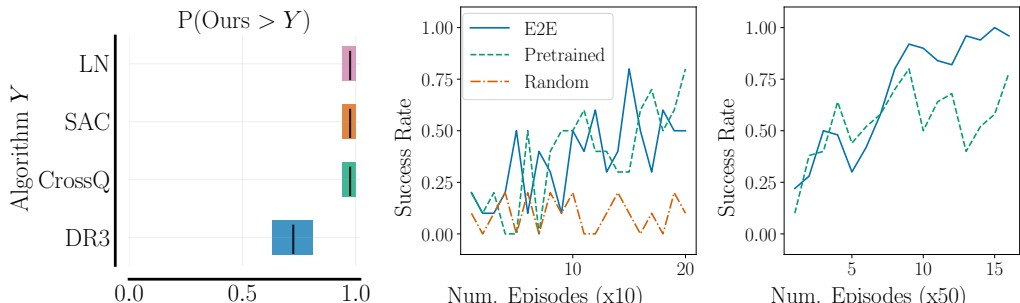

Figure 5: **(Left)** The probability of Simplified Q (Ours) being better than existing RL algorithms in success rate, run over three seeds. **(Middle)** Comparison between image encoders: (1) trained end-to-end (E2E), (2) randomly initialized image encoder, and (3) pretrained image encoder with HILP objective (Park et al., 2024). The model with a frozen randomly-initialized encoder fails to improve its grasp success rate even after online interactions, while the models trained end-to-end and with a frozen pretrained image-encoder can continually improve as it gathers more data. **(Right)** Asymptotic performance between training end-to-end and frozen pretrained image encoder. The model trained E2E appears to achieve better asymptotic performance than one with a frozen pretrained image encoder. All models are first pretrained for 100K gradient steps with 50 human-teleoperated demonstrations.

that leveraging a pretrained image encoder does improve upon using a frozen randomly-initialized network, but does not provide visible performance improvement over training E2E. In fact we have continually trained the Simplfied Q agents for 800 episodes and have observed that the E2E agent can eventually achieve above $90\%$ success rate, while the agent with pretrained image encoder only achieve up to near $80\%$ success rate (Figure 5, right). However, one benefit of using pretrained image-encoder is training less parameters, thereby reducing the learning-update time. In our experiments training on a fixed pretrained image-encoder takes approximately 14 seconds for performing learning updates between episodes while training E2E takes approximately 40 seconds.

### 4.3 LATENT FEATURE REPRESENTATION SIMILARITY AND Q-VALUE ESTIMATES

We now analyze the impact of our proposed regularizer. Our goal is to decorrelate the latent feature representation of different state-action pairs, which is measure by the dot product of the features $\Phi(s, a)^{\top} \Phi(s', a')$, where $(s, a)$ and $(s', a')$ are independently drawn from $\mathcal{D}$. We sample 512 random state-action pairs from a buffer of random trajectories and visualize their similarity in the feature space induced by each algorithm during offline RL and online RL. Figure 6a illustrates that using our proposed regularizer yields lower-magnitude dot product for most state-action pairs when compared to CrossQ and SAC. We can also observe a general trend that the magnitude decreases for all algorithms as the agent collects more data. We also visualize their Q-values to investigate whether the Q-function suffers from overestimation. We observe in Figure 6b that across 200 online interactions, the Q-values of Simplified Q is consistently below the maximum realizable returns, whereas CrossQ and SAC fail to achieve this. However, there is an interesting trend that SAC appears to begin with reasonable Q-values after the offline phase but quickly diverges during the online phase, while CrossQ has already diverged Q-values after the offline phase. Secondly, we notice another trend that the Q-values are decreasing in magnitude as the agents continually train the Q-function with CrossQ and SAC. We leave the investigation of this phenomenon for future work.

## 5 RELATED WORK

Reinforcement learning (RL) algorithms require extensive exploration to learn a performant policy, which is an undesirable property for robotic manipulation. Offline-to-online (O2O) RL is an alternative paradigm that also includes policy pretraining with offline data prior to online interactions, with the hope that the pretrained policy is satisficing (Song et al., 2023; Tan & Xu, 2024; Huang et al., 2024; Zhang et al., 2022; Li et al., 2023). O2O RL is also related to reinforcement learning

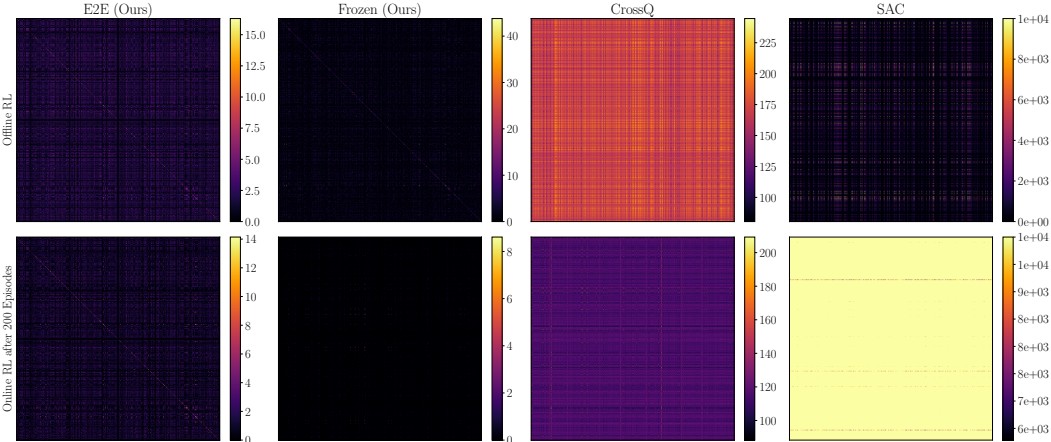

(a) The similarity between the latent representations, $\min(\Phi(s,a)^\top \Phi(s',a'), 10000)$, of 512 random state-action pairs after offline RL **(Top)** and 200 episodes of online RL **(Bottom)**. Simplified Q is able to maintain dot-products with small magnitude between different state-action pairs, indicating that the model can decorrelate these state-action pairs, whereas both CrossQ and SAC exhibit significantly larger magnitude dot-products, indicating that these models strongly correlates these different state-action pairs.

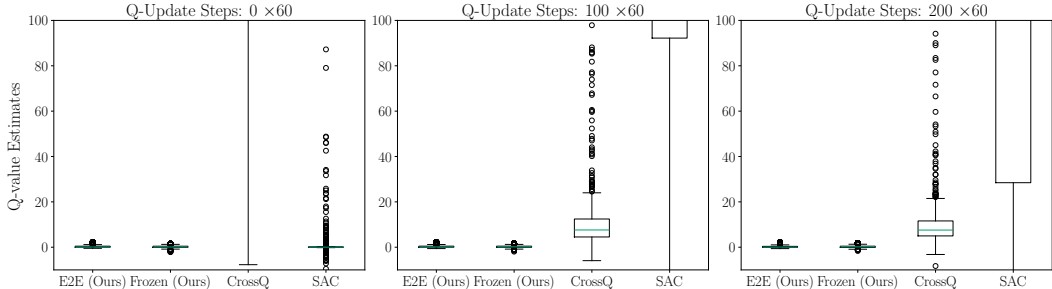

(b) The Q-value estimates of 512 random state-action pairs, evaluated at varying number of gradient updates on the Q-function. Simplified Q maintains maintains reasonable Q-values, while both CrossQ and SAC seem to have diverged in Q-values.

Figure 6: (a) The similarity between the latent representation of different state-action pairs. (b) The Q-value estimates of different state-action pairs. From left to right: Model trained end-to-end with Simplified Q (Ours). Model trained with Simplified Q with a frozen pretrained image encoder. Model trained with CrossQ with a frozen pretrained image encoder. Model trained with SAC with a frozen pretrained image encoder.

from demonstrations (RLfD) where the RL agents is equipped with demonstrations that are often assumed to be of high quality (Rajeswaran et al., 2017; Vecerik et al., 2017; Hester et al., 2018; Nair et al., 2018). Existing algorithms including DQfD (Hester et al., 2018), COG (Singh et al., 2020), TD3-BC (Fujimoto & Gu, 2021), Cal-QL (Nakamoto et al., 2023), and RLPD (Ball et al., 2023) have shown some successes in both simulated and real-life environments. While our work draws inspiration from these works, our approach differs in how we learn the Q-function. Though, our proposed method is similar to RLPD where the training is purely RL based without any behavioural-cloning objectives during policy learning. That means that our approach is agnostic to the quality of the data.

The offline phase of O2O RL has recently been an emphasis in the field (Kumar et al., 2020; Yu et al., 2020; Fujimoto et al., 2019). While there has been enormous successes in state-based domains, Lu et al. (2023) and Rafailov et al. (2024) have identified that these offline RL algorithms face additional challenges in the image-based domains. On the other hand, researchers in online RL have developed algorithms to account for image-based domains (Hansen et al., 2023; Li et al., 2022; Wang et al., 2022; Laskin et al., 2020; Yarats et al., 2021; Cetin et al., 2022; Parisi et al., 2022; Hu et al., 2024), particularly leveraging data augmentation and pretrained vision backbones. While we

consider leveraging pretrained vision backbone, our work demonstrates that this step is unnecessary, but it may accelerate learning in wall time. We also utilize data augmentation on the images only in the offline phase which enables faster training in wall time.

Online RL for real-life robotic manipulation tasks often involve sim-to-real transfer (Han et al., 2023; Tang et al., 2024). Though, there are several works that directly deploy online RL algorithms on real-life systems (Seo et al., 2024; Hertweck et al., 2020; Lampe et al., 2024; Luo et al., 2024). To address the exploration challenges, Hertweck et al. (2020) and Lampe et al. (2024) leverage hierarchical RL to decompose the task into human-interpretable behaviours. The hierarchical RL agent then explores the space in the temporally-abstracted MDP that allows better coverage. On the other hand, Seo et al. (2024) and Luo et al. (2024) leverage offline data to provide indirect guidance to the policy. Our work is similar to the latter line of research but also consider the pretraining phase to accelerate online learning.

Finally, our work is most relevant to the recent breakthrough on analyzing Q-learning through the lens of neural tangent kernel (NTK) (Jacot et al., 2018). Specifically, researchers have identified that the learning dynamics of Q-functions through NTK might describe why Q-learning tends to be unstable and diverge (Kumar et al., 2022; Yue et al., 2023; Ma et al., 2023; Tang & Berseth, 2024). Consequently the researchers proposed various regularization techniques (He et al., 2024) and model architectures (Yang et al., 2022) to alleviate this problem. We differentiate ourselves by identifying that the target network in (deep) Q-learning is often applied for similar reasons—decorrelating the consecutive state-action pairs during update. In particular we found that this target network is unnecessary when our regularizer is applied to decorrelate latent representation of state-action pairs.

## 6 CONCLUSION

In this work we introduced a novel regularizer inspired by the neural tangent kernel (NTK) literature that can alleviate Q-value divergence. We showed that this NTK regularizer can indeed decorrelate the latent representation of different state-action pairs, as well as maintaining reasonable Q-value estimates. Consequently we removed the target network altogether and replaced it with this NTK regularizer, resulting in our offline-to-online reinforcement learning algorithm, Simplified Q. We conducted experiments on a real-life image-based robotic manipulation task, showing that Simplified Q could achieve satisficing grasping behaviour immediately after the offline RL phase and further achieve above $90\%$ grasp success rate under two hours of interaction time. We further demonstrated that Simplified Q could outperform existing reinforcement learning algorithms and behavioural cloning with similar amount of total data. These results suggest that we can use purely reinforcement learning algorithms, without any behavioural-cloning objectives, for both offline and online training once we mitigate Q-value divergence. Finally, we should further reconsider whether we truly need to gather large amount of demonstrations for learning near-optimal policies in robotic applications. Additional discussions on limitations and future work are in Appendix A.

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

## A  LIMITATIONS AND FUTURE WORK

In the future we aim to demonstrate the robustness of Simplified Q through conducting experiments in other manipulation tasks and other domains, particularly longer-horizon tasks. Secondly, running reinforcement learning algorithms in real life still requires human intervention to reset the environment that may discourage practitioners from applying these algorithms. This limitation might be addressed through reset-free reinforcement learning (Gupta et al., 2021). Furthermore, the current learning updates are performed in a serial manner, a natural direction is to parallelize this such that the policy execution and policy update are done asynchronously (Mahmood et al., 2018; Wang et al., 2023).

Experimentally we have observed that Simplified Q can still diverge (not necessarily the Q-values) when trained with higher update-to-data (UTD) ratio. Indeed, UTD ratio has been a challenge (D'Oro et al., 2022). We have also observed that the dormant neuron problem is still prevalent (Sokar et al., 2023). We expect algorithms such as LOMPO (Rafailov et al., 2021), REDQ (Chen et al., 2021), and DroQ (Hiraoka et al., 2022b), possibly in combination with our regularizer, can leverage higher UTD ratio to further improve sample efficiency. Our work also takes advantage of using offline data that includes successful attempts to workaround the exploration problem. One question is to investigate whether we can include play data (Ablett et al., 2023) or data collected from other tasks (Chan et al., 2023)—leveraging multitask data can potentially enable general-purpose manipulation. It is still of interest to perform efficient and safe exploration—for example using vision-language models and/or constrained reinforcement learning algorithms to impose safe policy behaviour and goal generation (García & Fernández, 2015; Yang et al., 2021).

There remains a big gap between the theoretical understanding and the empirical results of Simplified Q. Particularly we hope to show that this regularization can bound the Q-estimates to be within realizable returns with high probability, and guarantee convergence of the true Q-function. This may involve analyzing the learning dynamics of temporal difference learning with our regularizer (Kumar et al., 2022; Yue et al., 2023). It will also be interesting to analyze the differences in the learning dynamics with image-based and state-based observations (Pope et al., 2021). An alternative theoretical question is to investigate whether Q-overestimation is truly problematic for policy learning (Mei et al., 2023).

## B  THE CHALLENGES IN VISUAL-BASED OFFLINE RL

In this section we reaffirm the difficulty of offline RL in image-based tasks, which is first observed in Lu et al. (2023) and Rafailov et al. (2024). We conduct an experiment on a real-life 2D reaching task using the UR10e arm. We define a state-based observation variant and an image-based observation variant to compare. The former accepts the same proprioceptive information specified in Section 4, but replacing the image with a delta target position. The latter leverages the same information, but the image is an arrow that indicates the location of the target relative to the current TCP position. The arrow's magnitude and angle correspond to the distance and direction respectively. We collect 50 demonstrations that are generated using a linear-gain controller, i.e. $\pi(x_{curr}, x_{goal}; K) = \text{clip}(K(x_{goal} - x_{curr}), -1, 1)$, where $K = 2.0$ is the gain parameter.

Figure 7, left, demonstrates that CQL, a state-of-the-art offline RL algorithm, can recover performance similar to the demonstrations in state-based reacher. Including data augmentation can further improve its performance. However, we observe that the exact same algorithm fails to recover the performance on image-based reacher (Figure 7, right). Even with data augmentation the offline RL agent is unable to recover the performance. On the other hand, BC can recover similar performance for both state-based and image-based observations.

## C  EXTRA EXPERIMENTAL RESULTS

### C.1  TRAJECTORIES OVER TRAINING

Here we provide few online interaction trajectories for each RL algorithm during training (Figure 8). Our Simplified Q agent can consistently go inside the bin, towards the item, and attempt to pick the item. On the other hand, both CrossQ and SAC have diverged in its behaviour during training.

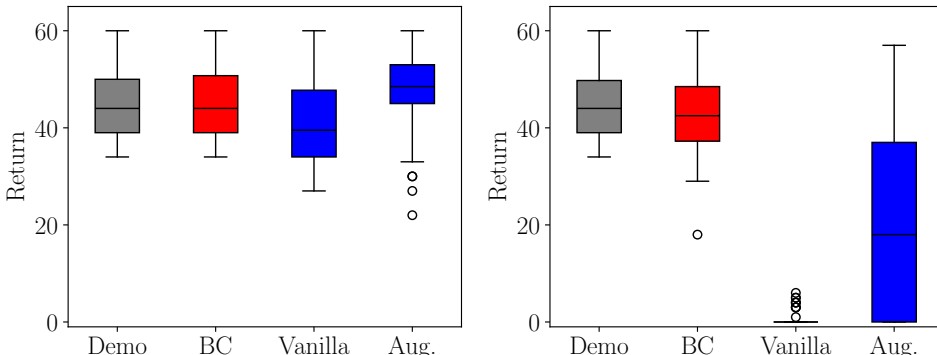

Figure 7: The returns of offline pretrained models on real-life 2D reacher environment, evaluated on 50 trials. Gray bar corresponds to demonstration data, red bar corresponds to behavioural cloning (BC), and blue bars correspond to offline RL, implemented with conservative Q-learning (CQL). Vanilla corresponds to no data augmentation. **(Left)** State-based observation. **(Right)** Image-based observation. Policies trained with BC remain consistent in performance with both state-based and image-based observations while policies trained with CQL fails to achieve similar performance with image-based observations.

Table 2: Ablation on various techniques and hyperparameters applied in our main experiments. The results are aggregated over 200 online RL episodes and we show the overall success rate (SR) and overall P-stop rate (PR). Our default setting is bolded in text.

(a) Ablation on including symmetric sampling (SS) and self-imitation learning (SIL). Excluding any of these techniques result in a slightly worse policy in both success rate and P-stop rate.

| $\beta$ | SR | PR |
|---|---|---|
| **Both** | 36.5% | 6.5% |
| w/o SIL | 28.5% | 16.0% |
| w/o SS | 31.5% | 17.5% |

(b) Ablation on $N$-step temporal-difference learning. It appears that when $N = 3$ performs better than $N = 1$ and $N = 5$.

| $N$ | SR | PR |
|---|---|---|
| 1 | 19.0% | 25.0% |
| **3** | 36.5% | 6.5% |
| 5 | 26.5% | 15.5% |

(c) Sensitivity analysis on the coefficient of our proposed regularizer $\beta$. We find $\beta \in [0.1, 0.4]$ to obtain consistent performance.

| $\beta$ | SR | PR |
|---|---|---|
| 0.0 | 22.5% | 25.0% |
| 0.1 | 54.0% | 3.5% |
| **0.2** | 36.5% | 6.5% |
| 0.4 | 44.0% | 16.0% |
| 1.0 | 22.0% | 0.5% |

## C.2 ABLATIONS

We investigate the importance of adding successful episodes to the demonstration buffer over time and symmetric sampling introduced by RLPD (Ball et al., 2023). We conduct an experiment where we run our approach with and without self-imitation learning (SIL), and without symmetric sampling (SS). Excluding any of SIL or SS resulted in an increase on the P-stop rate which suggests that our proposed technique can be safer as it enforces the agent to sample more positive samples as training progresses (Table 2a). We observe that the overall success rate is higher with our method, up to 8% improvement. We also evaluate the importance of $N$-step temporal-difference learning, with $N = \{1, 3, 5\}$ (Table 2b). Similar to Seo et al. (2024) we found that setting $N = 3$ performs the best—notably when $N = 1$ the agent performance is the worst. It is likely due to higher bias compared to larger $N$.

Finally, we conduct a sensitivity analysis on the coefficient of our proposed regularizer $\beta \in \{0.0, 0.1, 0.2, 0.4, 1.0\}$. Table 2c shows that our proposed regularizer is generally robust with coefficients $\beta \in [0.1, 0.4]$. Notably we found that when $\beta = 0.1$ both its cumulative success rate and P-stop rate to be superior than our chosen default, $\beta = 0.2$. Behaviourally, when $\beta = 0.0$ or $\beta = 1.0$ we observe that the policy can only pick from a very small region of item locations. The latter is worse as it can only pick towards the center of the bin.

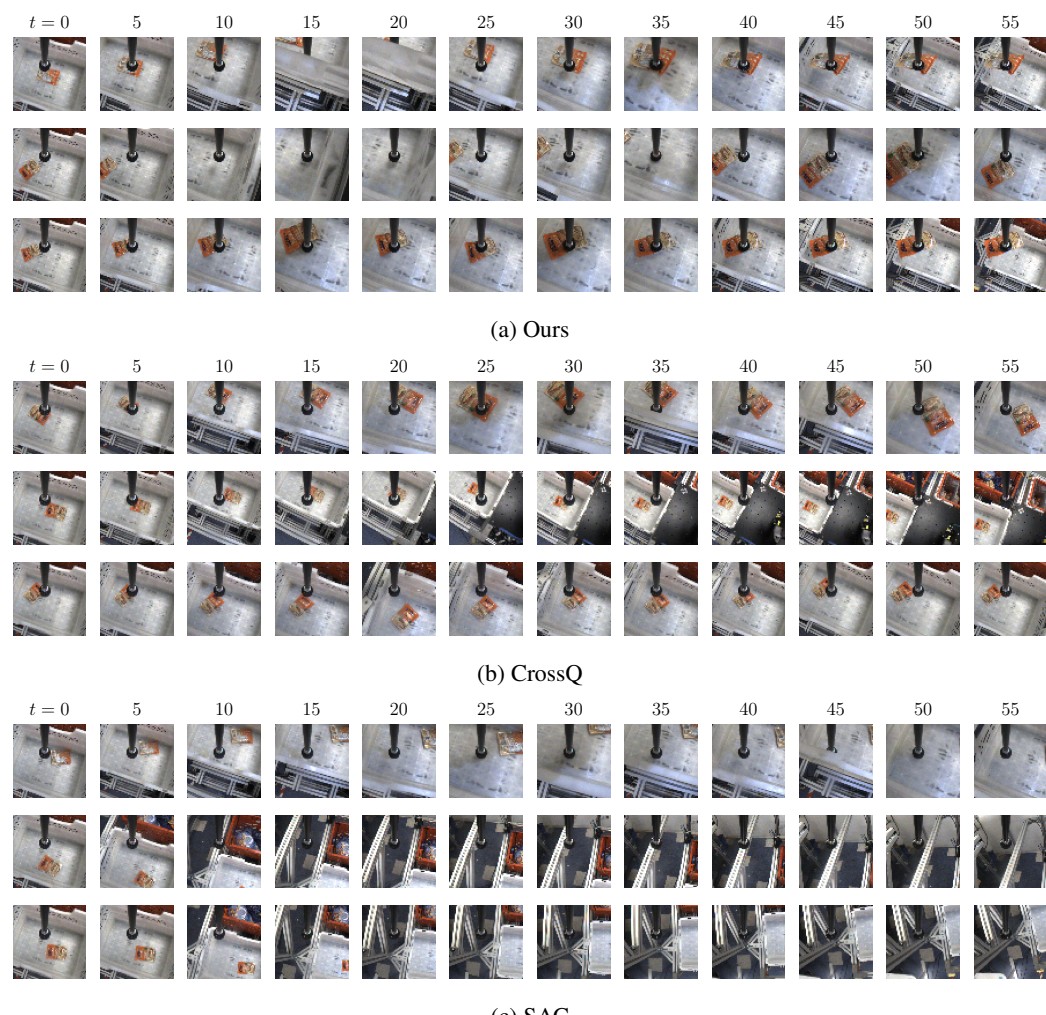

Figure 8: The $n$'th online interaction trajectory where $n = \{1, 50, 150\}$, from top row to bottom row, between three RL algorithms. Our algorithm can consistently go inside the bin and attempt to pick the item, while CrossQ and SAC have diverged during training.

## C.3 ZERO-SHOT GENERALIZATION

We now investigate the robustness of the RL policy trained with our proposed method without further parameter updates. We evaluate the agent on two other item types (Figure 9, right), one with different colour and one with different shape and rigidity. We also evaluate the agent on other scenarios where there are three items in the bin simultaneously and where the lighting condition changes. We evaluate each scenario for 50 grasp attempts. We emphasize that the agent has only seen a single item over the training runs, making these scenarios totally out-of-distribution (OOD).

Our result is shown in Figure 9, left, and we can see that the agent does degrade in performance under OOD scenarios, but we observe that the success rate is around 70% on the different coloured item, while achieving around 50% success rate on the item with different shape and rigidity. The latter fails more frequently as the item is significantly taller, resulting in the agent P-stopping more frequently due to unseen item height. Furthermore, the agent also degrades in performance when there are multiple items in the bin, and we observe that the main failure mode is when the gripper is in between two items at equidistance, which causes the policy to undercommit on one of the two items. This degradation is more significant when all the items are OOD. Finally, the modified lighting condition does cause a visible performance degradation on the policy, we suspect this is

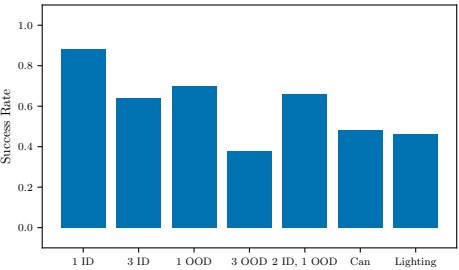 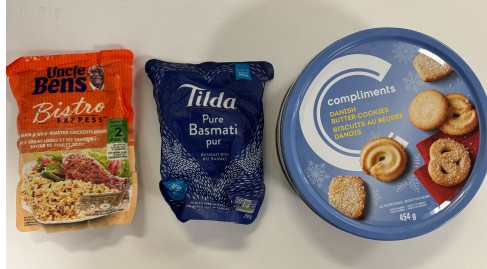

Figure 9: **(Left)** Zero-shot generalization on unseen scenarios using our trained RL policy. We compute the success rate on 50 attempts for each evaluation. The number corresponds to the item count in the bin. *(ID)* In-distribution item: orange rice bag. *(OOD)* Out-of-distribution item: blue rice bag. *(Can)* Out-of-distribution item: cookie can. *(Lighting)* Out-of-distribution lighting condition: Different light source. **(Right)** The item roster for zero-shot evaluation. The trained policy is still able to pick under various OOD scenarios. However including multiple different-coloured items, different item type, and different lighting condition can significant degrade the grasp success rate.

related to the light reflection on the item. Particularly, the policy cannot differentiate between the bottom of the bin and the reflection of the item, thereby neglecting the item completely.

## D    VISUALIZING THE IMAGE LATENT REPRESENTATION

We now inspect the latent representations induced by each image encoder in Figure 10—namely pretrained image encoder with successive representation **(HILP)**, randomly-initialized image encoder **(Random Init.)**, end-to-end trained image encoder **(E2E)** of the Q-function immediately after offline RL **(Offline)**, 200 online RL interactions **(200 Episodes)**, and 800 online RL interactions **(800 Episodes)**. We project the 64-dimensional latent representation onto a 3-dimensional space using UMAP (McInnes et al., 2018). We observe that HILP can nicely separate trajectories—each strand displays a smooth colour gradient from the first timestep to the last timestep (Figure 10, top). The strands can also describe the motion of the arm through a grasp attempt, particularly linear $z$ velocity is extremely negative as the arm reaches toward the item inside the bin and becomes extremely positive once the arm acquires a vacuum seal on said item. On the other hand, randomly initializing the image encoder fails to achieve the same, furthermore it cannot separate images that have dramatically different linear $z$ velocity (Figure 10, bottom). The representation induced by offline RL is similar to one induced by the random-initialized image encoder, but we can visually see that the images corresponding to positive linear $z$ velocity are clustered together which suggests that the encoder can identify when the item has been grasped. This representation is further refined as the image encoder is updated with more online interaction data, clearly separating the images with different linear $z$ velocity actions, and also induces a smoother colour gradient in terms of timesteps.

## E    HYPERPARAMETERS AND ALGORITHMIC DETAILS

The self-imitation technique is inspired by soft-Q imitation learning (Reddy et al., 2020) and RLPD (Ball et al., 2023), where the algorithm samples transitions symmetrically from both the interaction buffer $\mathcal{D}_{\text{on}}$ and the offline buffer $\mathcal{D}_{\text{off}}$. However, symmetric sampling samples half of the transitions from the offline data $\mathcal{D}_{\text{off}}$, which is undesirable when the average return induced by $\mathcal{D}_{\text{off}}$ is lower than the current policy $\pi$. We therefore include successful online interaction episodes into $\mathcal{D}_{\text{off}}$ in addition to the interaction buffer $\mathcal{D}_{\text{on}}$, a technique inspired by self-imitation learning (Oh et al., 2018; Seo et al., 2024). The consequence is twofold: (1) it allows the agent to see more diverse positive examples as it succeeds more, and (2) this results in the buffers being closer to on-policy data as the current policy becomes near optimal.

We choose the CQL regularizer coefficient to be $\alpha = 1.0$ and the NTK regularizer coefficient to be $\beta = 0.2$. We use Adam optimizer (Kingma, 2014) with learning rate 0.0003 for offline training and end-to-end online RL; we use a smaller learning rate 0.00005 for online RL with frozen image encoder as we found that using same learning rate is less stable. The batch size is set to be 512

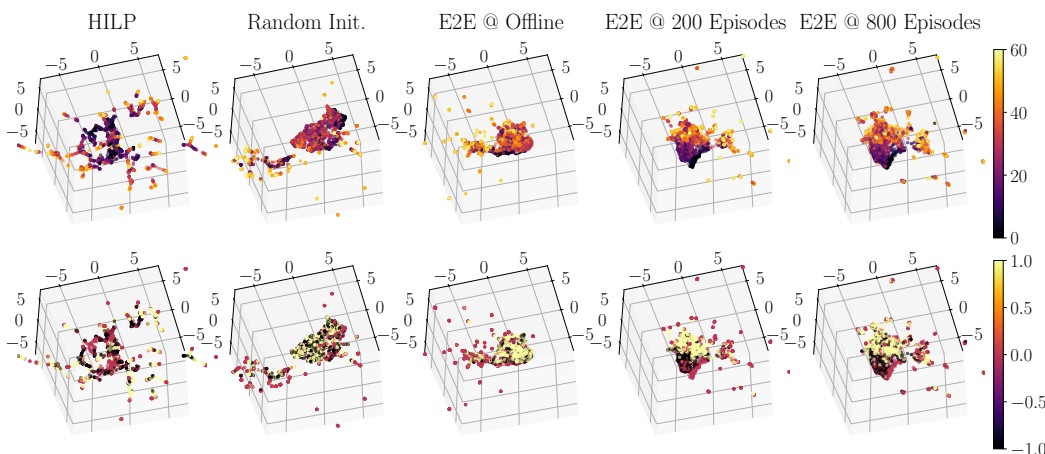

Figure 10: The latent representation induced by different image encoders, evaluated at different training phase on the same demonstration data. **(Top)** The points are colour-coded by the timestep. **(Bottom)** The points are colour-coded by linear $z$ velocity action. The pretrained image encoder with HILP can provides a timestep-interpretable representation, where each strand corresponds to a particular trajectory. The linear $z$ velocity can also be nicely interpreted—its value is extremely negative as the arm reaches toward the item and extremely positive as the arm lifts the item. randomly-initialized encoder fails to separate images into interpretable representation. The image encoder trained end-to-end (E2E) slowly clusters images with similar timesteps and linear $z$ velocity as online RL continues.

(i.e. sampling 256 from $\mathcal{D}_{\text{off}}$ and 256 from $\mathcal{D}_{\text{on}}$ during online phase), and we perform 60 gradient updates between every attempt for both the policy and Q-functions to avoid jerky motions. The models update at $\geq 1$ update-to-data ratio—when P-stop occurs the ratio is higher due to shorter trajectory length.

The image encoder is a ResNet (He et al., 2016), followed by a 2-layer MLP. For the policy, the MLP uses `tanh` activation with 64 hidden units, whereas for the Q-function, the MLP uses `ReLU` activation with 2048 hidden units. Training with a frozen image encoder spends approximately 14 seconds, whereas training end-to-end (E2E) spends approximately 40 seconds. This discrepency comes from updating less parameters for the former—we note that in E2E each model has its own separate image encoder. To pretrain the image encoder, we use first-occupancy Hilbert representation objective introduced by Moskovitz et al. (2022) and Park et al. (2024). The image encoder is trained with the images from same 50 demonstrations for 50K gradient steps.

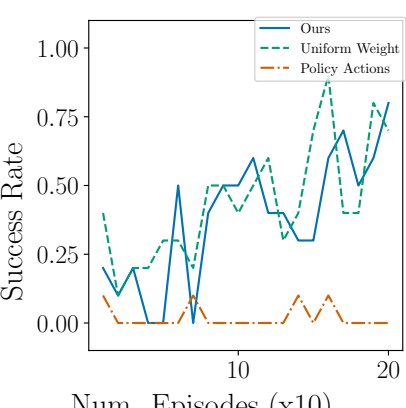

Figure 11: Comparing our method with using policy action for the CQL regularizer and applying uniform-weighting on the CQL regularizer. Using policy action significantly degrades the performance of Simplified Q.

For CQL, we found that setting $\mu = \mathcal{U}(\mathcal{A})$ instead of $\mu = \pi$ to be more effective (Figure 11). We further visualized the gradient field of the Q-function w.r.t. actions and observe that using $\mu = \pi$ tends to cause the learner policy to get stuck in a local optimum, whereas sampling from $\mathcal{U}(\mathcal{A})$ allows for a smooth landscape with less local optima (Figure 12). Finally, rather than pushing Q-values of in-distribution actions towards infinity in the CQL regularizer, we apply a weighted penalty based on the distance between the in-distribution action and the action sampled from $\mu$. Specifically, the CQL regularizer is implemented as

$$\mathbb{E}_{\mathcal{D},\mu}\left[\left(1 - \exp(-\|a - a'\|^2)\right) Q_\theta(s, a')\right],$$

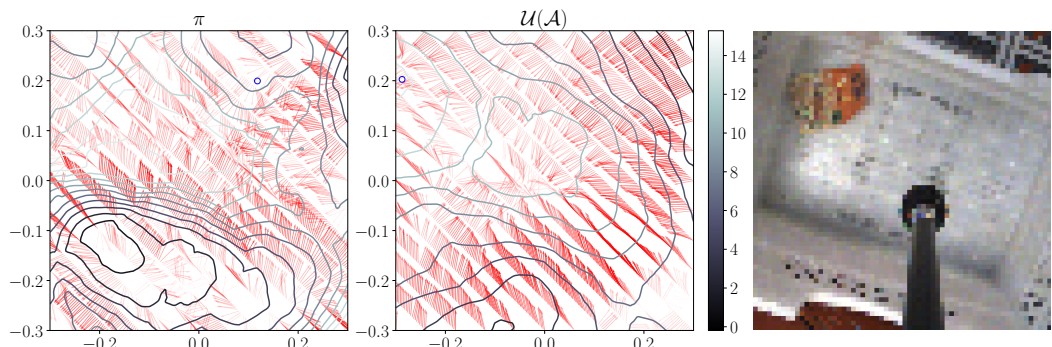

Figure 12: The gradient of $Q$ w.r.t. linear $xy$ velocities and the corresponding image observation. **(Left)** The Q-function is learned through setting $\mu = \pi$. **(Middle)** The Q-function is learned through setting $\mu = \mathcal{U}(\mathcal{A})$. **(Right)** The corresponding image observation. The opacity of the arrow corresponds to the magnitude and the contour corresponds to the Q-values. The blue circle corresponds to the policy's predicted action. We observe that with $\mu = \mathcal{U}(\mathcal{A})$ the gradient field tends to point towards the direction of the item whereas $\mu = \pi$ seems to have multiple local optima on different directions.

where $a \sim \mathcal{D}$ is the in-distribution action and $a' \sim \mu$ is the (possibly) out-of-distribution action.

For BC, we use the exact same policy architecture, learning rate, optimizer, batch size, and number of gradient steps as the RL counterpart, and perform maximum likelihood estimation which corresponds to minimizing the mean-squared error:

$$\mathcal{L}_{BC}(\pi) = \frac{1}{|\mathcal{D}_{\text{off}}|} \sum_{(s,a) \in \mathcal{D}_{\text{off}}} \|a - \pi(s)\|^2.$$

While one may use validation loss to select the "best" policy, we evaluate each checkpoint at every 10K gradient steps and have observed that the performance does not vary significantly, corroborating previous findings (Ablett et al., 2023; Hussenot et al., 2021; Mandlekar et al., 2022).

