# OpenReview forum: "Offline-to-online Reinforcement Learning for Image-based Grasping with Scarce Demonstrations"
_ICLR.cc/2025/Conference — Submitted to ICLR 2025_

### Official Review · Reviewer_awMg · 2024-10-21

**Soundness:** 2
**Presentation:** 3
**Contribution:** 2
**Rating:** 5
**Confidence:** 3

**Summary:**

This paper proposes a regularization method for CQL, improving efficiency in the offline-to-online RL setting. Inspired by the neural tangent kernel (NTK) literature, the paper replaces the target Q network with an NTK regularization term. Experiments are conducted on an image-based grasping task using a vacuum. Results show that the method outperforms BC and RL baselines after online RL and does not rely on a pre-trained image encoder.

**Strengths:**

1. To my knowledge, the proposed NTK regularization for offline-to-online CQL is a novel approach that addresses the learning stability issues.
2. The paper demonstrates promising experimental settings on a real-world robotic grasping task, taking acceptable training time to improve the policy on the real robot.
3. The paper presents a detailed experimental analysis of the grasping task.

**Weaknesses:**

1. My main concern is that the proposed method appears to be a general approach for offline-to-online RL rather than designed for robotic grasping. The motivation behind the proposed regularization term does not seem related to the task of robotic grasping. However, the majority of the paper is written as if it is focused on robotic grasping.

2. Because of this, I suggest that the experiments should include various RL benchmarks, such as D4RL, to demonstrate the general effectiveness of the proposed algorithm, rather than focusing solely on a single vacuum grasping task.

3. Regarding the literature on robotic grasping, the use of a vacuum to grasp flat objects, as demonstrated in the paper, represents a relatively simple setup. It remains unclear whether the method would be effective with parallel grippers or in grasping more complex objects. If hardware devices are not available, conducting simulation experiments on these settings could strengthen the experimental results.

**Questions:**

Please see the Weaknesses.

---

> ### Author Response · Authors · 2024-11-18
>
> We thank reviewer awMg for the thoughtful comments. We are grateful to see that the reviewer finds our work novel and finds our real-life experiments and analyses as strengths. Below we address their comments and questions and we hope they can reconsider our score:
>
> > 1. Concern on having a more general approach for O2ORL rather than robotic grasping
>
> We thank reviewer awMg for raising this concern. We agree that our method does not explicitly leverage properties of robotic grasping, e.g., invariance, equivariance, or some other inductive biases. Our work is inspired by the real-life challenges encountered in robotic manipulation—we address the question of why there is so limited real-life robotic research done with reinforcement learning rather than imitation learning. Real-life robotic manipulation tasks have very unique properties that other common RL benchmarks fail to capture, e.g., sparse reward and challenging exploration in continuous action space. As shown in D5RL (Rafailov et al., 2024), the image-based tasks are very challenging and have not been explored widely in the RL literature.
> We also appreciate that reviewer awMg has pointed out that our approach is agnostic to robotics—we believe that a more general approach than imposing domain-specific knowledge enables better applicability and adoption.
>
> >  2. More benchmarks to show effectiveness.
>
> This is a fair concern. To address reviewer awMg’s concern on a wider range of tasks, we conduct offline RL experiments in robomimic (Table 1), also suggested by reviewer cV3D. Based on the simulated results we can observe that Simplified Q is better than CQL on majority of the tasks, but similar to our original findings none of the offline RL algorithms is competitive against BC.
>
> > 3. Use of vacuum to grasp being relatively simple
>
> We agree with reviewer awMg that using parallel grippers will be more challenging. However, our item can easily crumble and deform, making it difficult to create a vacuum seal at times—the agent will need to identify smoother areas to increase the chance of a vacuum seal. We further demonstrate that four other existing algorithms fail to successfully grasp the items (Figures 2 and 3), perhaps this suggests that the problem is not as easy as it seems. Finally, we hope our simulation experiments in the previous response address this concern as well.
>
> References:
>
> Rafael Rafailov, Kyle Beltran Hatch, Anikait Singh, Aviral Kumar, Laura Smith, Ilya Kostrikov, Philippe Hansen-Estruch, Victor Kolev, Philip J. Ball, Jiajun Wu, Sergey Levine, and Chelsea Finn. D5RL: Diverse datasets for data-driven deep reinforcement learning. Reinforcement Learning Journal, 5:2178–2197, 2024.

---

> ### Author Response · Authors · 2024-11-27
> **Gentle reminder**
>
> Dear Reviewer awMg,
>
> We thank you for the time spent reviewing our submission.
>
> As the main discussion phase is ending soon, we wanted to send this gentle reminder. We have done our best to answer the comments you raised, as well as incorporate your suggestions. We would love to hear back from the reviewer and whether we have addressed their concerns.

---

### Official Review · Reviewer_L1Gj · 2024-10-30

**Soundness:** 3
**Presentation:** 3
**Contribution:** 1
**Rating:** 5
**Confidence:** 4

**Summary:**

The paper proposes a variation on conservative Q-learning (CQL) to use when training a robot grasping policy in a low data regime.

Drawing inspiraction from the Neural Tangent Kernel line of work, the Q-function is parameterized as $Q(s,a) = w^T \phi(s,a)$ where $w$ are the weights at the last layer and $\phi(s,a)$ is some (potentially deep net based) featurizer. If we froze the weights of $\phi(s,a)$, then the kernel $\kappa(s,a,s',a')$ would be exactly $\phi(s',a')^T\phi(s,a)$, and this controls the amount of generalization occurring between state action pairs.

In this work, we do not freeze $\phi(s,a)$, but instead add an auxiliary loss to CQL to keep $\phi(s',a')^T\phi(s,a)$ close to 0 to reduce overgeneralization (a key challenge in offline RL). This is done with an mean squared loss, $E_D[ (\phi(s, a_u)^T \phi(s', a_\pi'))^2]$, encouraging decorrelation between a uniformly sampled action $a_u$ and the current policy's action $a_\pi$.

This method is studied in a real-life grasping task.

**Strengths:**

The paper does a good job of presenting the core change of their method, as well as motivating why it could help with overgeneralization. The model architectures used are well-defined

**Weaknesses:**

The method doesn't seem to meaningfully build on prior work.

* RL and BC have been applied to image-based grasping before, including in the low demonstration regime.
* Image-based RL has on occasionally been done before as well (with SAC)

The absolute performance numbers relative to the BC baseline seem quite low. The paper is correct in observing that the NTK objective from Kumar et al 2022 is partly ill-formed: minimizing $E_D[ \phi(s, a_u)^T \phi(s', a_\pi')]$ allows for a negative kernel value, which could let learning in one Q-value cause unlearning in another Q-value. However, the Ma et al 2023 citation is described as also doing this, which is incorrect. The Ma et al 2023 paper minimizes the absolute value $|\phi(s, a_u)^T \phi(s', a_\pi')|$, i.e. it uses an L1 loss. This paper does technically do something different (L2 loss instead), but the results of doing so aren't great.

In short I'm not very convinced on novelty or effectiveness. This combined with using a single hard-to-reproduce robot environment make me believe the paper is not good enough.

**Questions:**

Could the authors explain why they remove the target network when including the NTK regularizer? As far as I can tell, the target network is maintained in prior work that uses NTK regularization, I am not sure why it was removed.

---

> ### Author Response · Authors · 2024-11-18
>
> We thank reviewer L1Gj for the detailed feedback. We are grateful to see that reviewer L1Gj finds our work to be well motivated and well presented. Below we address their comments and questions and we hope they can reconsider our score:
>
> > Not meaningfully build on prior work.
>
> While we agree with reviewer L1Gj that there are image-based manipulation tasks in previous work, we are not aware of any O2ORL that uses purely RL objectives (i.e. no BC regularization term). We also agree that DrQ and follow-up work have worked on MuJoCo continuous-control tasks with success. However, these environments are often in the dense-reward setting. Existing image-based RL algorithms that appear to work in the sparse-reward setting, as far as we know, are ones included in our related work such as SAC-X (Hertweck et al., 2020) and CQN (Seo et al., 2024)---the latter also uses BC regularizer and appears to fail without it. We are happy to include any extra references if reviewer L1Gj knows of any new result.
>
> > The absolute performance numbers relative to the BC baseline seem quite low.
>
> If we understand correctly, we believe reviewer L1Gj refers to offline RL performing worse than BC baseline. We agree with this observation, but we note that in the online phase, the Simplified Q eventually outperforms BC with half of the total data (50 demonstration episodes + 200 interaction episodes vs. 500 demonstration episodes). This result is consequential as we no longer require human teleoperators to gather a large amount of demonstrations, which can be costly in both time and resources.
>
> > The Ma et al 2023 paper uses an L1 loss… Not very convinced on novelty or effectiveness and single hard-to-reproduce robot environment.
>
> We thank reviewer L1Gj for pointing this out, indeed we had made a mistake on referring to this paper in the statement. We have removed the reference in the updated manuscript. We highlight that both existing works are different to Simplified Q and still leverage the target network. Notably, DR3 uses consecutive state-action pairs as its implicit regularization is motivated differently from ours. Ma et al. (2023) introduces an extra OOD policy, which increases the number of parameters and hyperparameters. Our goal instead is to minimize Q-divergence and further simplify learning the Q-function through reducing previously mentioned quantities. We have added a paragraph, lines 198-203, in our updated manuscript to describe the differences.
>
> Our novelty lies in identifying that the Q-function learning dynamics with gradient descent can result in a simplification of Q-learning. Previous work ignores this possibility and introduces extra components by having an extra policy (Ma et al., 2023) or maintaining more copies of model parameters (Tang and Berseth, 2024). Our approach reduces the number of model parameters and the number of hyperparameters related the target network into a single scalar which is also easy to tune (see Table 2c). This enables practitioners to run the algorithm in real-life applications due to less hyperparameters to tune and potentially lessen the compute burden. As far as we know we are also one of the first to successfully perform O2O RL on real-life image-based manipulation, with purely RL objective without any BC regularizers, e.g., TD3-BC (Fujimoto and Gu, 2021) and CQN (Seo et al., 2024). Once again, we would be happy if reviewer L1Gj is aware of existing work that has already accomplished this.
>
> Regarding effectiveness, we thank the reviewer for pointing out the lack of experiments. We have increased the number of runs and included additional baselines to demonstrate the effectiveness of our approach in the real-life grasping task (Figures 3 and 4). Specifically we include two more algorithms, DR3 (Kumar et al., 2022) and SAC + LayerNorm (Yue et al., 2023), with three seeds per algorithm. The results remain consistent with our original claim although DR3 appears to perform better than other RL baselines except for Simplified Q. Finally, we conduct offline RL experiments in robomimic (Table 1), as suggested by reviewer cV3D and reviewer awMg. Based on the simulated results we can observe that Simplified Q is better than CQL on majority of the tasks, but similar to our original findings none of the offline RL algorithms is competitive against BC. We hope these extra results address the reviewer’s concern on the effectiveness as we add more statistical analyses and on more tasks.
>
> > Could the authors explain why they remove the target network when including the NTK regularizer?
>
> Thank you for asking this question. The target network is initially introduced to mitigate the temporal correlation between consecutive state-action pairs, which is already decorrelated using our regularizer. As a result, including the target network shows no obvious benefits anymore. The main benefits are reducing the amount of compute resources and ease of choosing hyperparameters.

---

> ### Author Response · Authors · 2024-11-18
>
> References:
>
> Younggyo Seo, Jafar Uruc¸, and Stephen James. Continuous control with coarse-to-fine reinforcement learning. In Conference on Robot Learning (CoRL), 2024.
>
> Tim Hertweck, Martin Riedmiller, Michael Bloesch, Jost Tobias Springenberg, Noah Siegel, Markus Wulfmeier, Roland Hafner, and Nicolas Heess. Simple sensor intentions for exploration. arXiv preprint arXiv:2005.07541, 2020.
>
> Aviral Kumar, Rishabh Agarwal, Tengyu Ma, Aaron Courville, George Tucker, and Sergey Levine. Dr3: Value-based deep reinforcement learning requires explicit regularization. In The Tenth International Conference on Learning Representations (ICLR), 2022.
>
> Yang Yue, Rui Lu, Bingyi Kang, Shiji Song, and Gao Huang. Understanding, predicting and better resolving q-value divergence in offline-rl. In Advances in Neural Information Processing Systems (NeurIPS), volume 36, pp. 60247–60277, 2023.

---

> > ### Comment · Reviewer_L1Gj · 2024-11-26
> > **Reply**
> >
> > Thanks to the authors for replying.
> >
> > > While we agree with reviewer L1Gj that there are image-based manipulation tasks in previous work, we are not aware of any O2ORL that uses purely RL objectives (i.e. no BC regularization term).
> >
> > There is also the QT-Opt line of work (https://arxiv.org/abs/1806.10293) which does offline Q-learning on images with only a Q-learning term and no BC regularization. As well as earlier work prior to DrQ on using autoencoders to stabilize SAC losses (https://arxiv.org/abs/1910.01741), although this one is more online in nature. For offline-to-online, there is also IQL (https://arxiv.org/abs/2110.06169) which is an offline method that has some experimental results on doing online finetuning afterwards. This was originally demoed on MuJoCo tasks but was later extended to image environments in future work (https://arxiv.org/html/2307.11949v4).
> >
> > > If we understand correctly, we believe reviewer L1Gj refers to offline RL performing worse than BC baseline. We agree with this observation, but we note that in the online phase, the Simplified Q eventually outperforms BC with half of the total data (50 demonstration episodes + 200 interaction episodes vs. 500 demonstration episodes). This result is consequential as we no longer require human teleoperators to gather a large amount of demonstrations, which can be costly in both time and resources.
> >
> > Thank you for making this clearer, it was not clear from the text. I still think it is a bit suspect to have an offline RL method perform worse than the BC baseline before online interaction (typically it is possible to at least tie the BC baseline), but it is good to see that online interaction can close the gap.
> >
> > So, to summarize, the main claims of separation to prior work are:
> >
> > * That in DR3 the loss is allowed to push towards negative correlation, rather than 0 correlation.
> > * That in the Ma 2023 work, they train an OOD policy concurrently with the current policy, and instead it is sufficient to use a uniformly random action as the OOD policy instead.
> >
> > I do think this is more of an improvement but I'm not sure it is a sufficient one, given that CQL uses a similar uniform-random-as-OOD setup when trying to be more conservative on unseen actions. I will adjust my rating upwards from 3 though.

---

> > > ### Author Response · Authors · 2024-11-27
> > >
> > > We thank reviewer L1Gj for the follow-up discussions and their thoughts.
> > >
> > > > QT-Opt, SAC-VAE, IQL, etc. and their performances
> > >
> > > Thank you for bringing these works to our attention. Indeed, we had missed QT-Opt which uses large amount of offline data (580K grasp attempts) with large amount of online data (26K online grasp attempts), in total of at least 800 robot hours. In comparison, our experiments take at most 3 robot hours per run. SAC-VAE is more related to online image-based RL on MuJoCo tasks which we emphasize to be dense-reward tasks.
> > >
> > > Regarding IQL, indeed it is a strong baseline but it is still an open question on whether IQL always outperforms CQL (e.g. Table 2 of [1] and D4RL Section of [2]). We can see in Table 2 of [1] that IQL is worse than CQL on Lift and Can, but better in Square. Generally BC also outperforms all other offline RL variants. This result corroborates our new Robomimic experimental results---we ran all the experiments in our table. Therefore, we do not believe that BC being better is suspicious in general given reasonable demonstrations.
> > >
> > > > Our contribution / novelty
> > >
> > > We emphasize that in addition to the listed points by the reviewer, our novel contribution comes from removing the target network which has been *critical for stabilizing RL algorithms*. We can now remove this target network due to the new observation of orthogonalizing the latent representation $\Phi(s, a)$ and $\Phi(s', a')$, for any $(s, a) \times (s', a')$, can mitigate Q-divergence. There are impactful consequences as we had previously mentioned, most notable one being removing the number of hyperparameters and model parameters, thereby improving the applicability of our method in real-life systems.
> > >
> > > References:
> > > [1] Jianlan Luo, Perry Dong, Jeffrey Wu, Aviral Kumar, Xinyang Geng, and Sergey Levine. "Action-quantized offline reinforcement learning for robotic skill learning." In Conference on Robot Learning, pp. 1348-1361. PMLR, 2023.
> > > [2] https://robomimic.github.io/docs/datasets/d4rl.html#d4rl-results

---

### Official Review · Reviewer_cV3D · 2024-11-03

**Soundness:** 2
**Presentation:** 2
**Contribution:** 3
**Rating:** 6
**Confidence:** 2

**Summary:**

This paper presents a novel regularizer inspired by neural tangent kernel (NTK) literature that can decorrelate the latent representation of different state-action pairs, while maintaining reasonable Q-value estimates. This is presented as a novel offline-to-online reinforcement learning algorithm to learn image-based tasks with small number of demonstrations. The proposed algorithm replaces the target network in off-policy actor-critic algorithms and is shown to overperform alternative methods.

**Strengths:**

The paper is structured fairly well and evaluations are carried out with experiments and comparisons that demonstrate the proposed algorithm.
The method presented in well framed in relevant literature and presented and formulated in a sound way.
The work is well motivated and its application is relevant to the field, and especially interesting for real-robot applications.
The experimental setup is explained well and plots and graphs are clear.
The paper analyses different aspects of the proposed method, eg latent feature, Q-estimates, performance against BC and SAC.

**Weaknesses:**

The clarity of the paper can be improved in terms of readability and crispness of contributing statements. For example in Figure 3, it is not immediately clear from the figures why the three plots show performance of different algorithms, while the caption is very verbose and sometime vague (eg "various image encoder" could be more specific).
While it is nice to see comparisons with BC and SAC, it would be interesting to see performance of conservative Q learning as well, as this method is based on it.
Other experiments would have helped support the contribution, eg experiments on D4RL, robomimic, or other available data/tasks.

**Questions:**

- In which cases does your method stop outperforming others? for example, is there a dependency on the number or quality of demonstrations?
- Can you clarify what aspect in particular of the NTK helps most? could other kernel be used/compared? if not, why?
- Did you test your algorithms on other baselines/tasks?

---

> ### Author Response · Authors · 2024-11-18
>
> We thank reviewer cV3D for the thoughtful feedback. We are grateful to see that reviewer cV3D finds our work and results are presented and formulated in a sound manner. We also appreciate that they recognize the importance of real-life robot applications. Below we address their comments and questions and we hope they can reconsider our score:
>
> **Weaknesses**
>
> > The clarity of the paper can be improved in terms of readability and crispness of contributing statements.
>
> We thank reviewer cV3D for the feedback on the clarity of our results. We have updated the caption of (previously Figure 3, now Figure 5) to specify the different image encoders. Specifically, we compare (1) trained end-to-end, (2) randomly initialized frozen image encoder, and (3) pretrained image encoder using HILP (Park et al., 2024).
>
> > ​​While it is nice to see comparisons with BC and SAC, it would be interesting to see performance of conservative Q learning as well, …
>
> We agree that we should include more experiments such as more baselines, environments, and tasks. Regarding the baseline comparisons, we apologize for potential confusion. Our compared RL baselines, including the newly added baselines, all include the CQL regularizer as we always pretrain the actor and critic. What is surprising here is that even with CQL the Q-values do end up diverging (Figure 6b).
> We again thank reviewer cV3D on suggesting further experiments on simulated environments and/or data. As an alternative we have conducted offline RL experiments in robomimic (Table 1), as suggested by you and reviewer awMg. Based on the results we can observe that Simplified Q is better than CQL on majority of the tasks, but similar to our original findings none of the offline RL algorithms is competitive against BC.
>
> **Questions**
>
> > In which cases does your method stop outperforming others? for example, is there a dependency on the number or quality of demonstrations?
>
> This is a great question, we have included relevant discussions in Appendix A but will further describe them here.
> Simplified Q can still diverge (not necessarily the Q-values) when trained with higher update-to-data (UTD) ratio. Indeed, UTD ratio has been a challenge (D’Oro et al., 2022). We have also observed that the dormant neuron problem is still prevalent (Sokar et al., 2023). We expect algorithms such as LOMPO (Rafailov et al., 2021), REDQ (Chen et al., 2021), and DroQ (Hiraoka et al., 2022), possibly in combination with our regularizer, can leverage higher UTD ratio to further improve sample efficiency.
>
> Another limitation comes from the exploration challenge in our environment. We have observed that none of the compared algorithms can grasp the item even with a larger number of episodes. We believe that more effective exploration can outperform our approach.
> Finally, we believe that using less data will result in narrower coverage of the dataset—in offline RL literature this appears to be a necessary condition (Zhan et al., 2022, Tkachuk et al., 2024). Unless we allow pretraining from multi-task datasets (Padalkar et al., 2023) or impose inductive biases such as equivariance (Tangri et al., 2024), it is unclear to us whether we can do any better.
>
> > Can you clarify what aspect in particular of the NTK helps most? could other kernel be used/compared? if not, why?
>
> We thank the reviewer for the questions. As suggested by our derivation in Equations 2 and 3, the influence of the current Q-update on any Q-values is dictated by the similarity (i.e. dot product) of the state-action pairs. The magnitude of the dot product positively correlates with the change of the Q-value on the other state-action pair. Therefore orthogonalizing (i.e. decoupling) the features will mitigate Q-value changes, at the cost of restricting generalization. On the other hand, this decoupling will mitigate the impact of using the Q-network itself for bootstrapping, thereby allowing us to drop the target network altogether.
>
> There are still other limitations however. First, this requires that the model itself is parameterized such that gradients can be computed—non-parameterized models like decision trees for example will no longer be applicable. Second, the theory on neural tangent kernels is still restricted to the infinite-width neural network setting despite aligned empirical observations. We believe that there are other necessary conditions for effective policy learning since SAC and other existing algorithms can still learn a good policy under Q-divergence. To that end perhaps other kernels can be developed due to the dependence of this policy learning component.
>
> > Did you test your algorithms on other baselines/tasks?
>
> We thank reviewer cV3D for this question. We have updated our manuscript to include more baseline algorithms including DR3 (Kumar et al., 2021) and SAC + LayerNorm (Yue et al., 2023). As mentioned in the previous response, we have also conducted offline RL experiments in robomimic (Table 1).

---

> ### Author Response · Authors · 2024-11-18
>
> References:
>
> Seohong Park, Tobias Kreiman, and Sergey Levine. Foundation policies with hilbert representations. In International Conference on Machine Learning (ICML), 2024.
>
> Pierluca D'Oro, Max Schwarzer, Evgenii Nikishin, Pierre-Luc Bacon, Marc G. Bellemare, and Aaron Courville. Sample-efficient reinforcement learning by breaking the replay ratio barrier. In Deep Reinforcement Learning Workshop NeurIPS, 2022.
>
> Ghada Sokar, Rishabh Agarwal, Pablo Samuel Castro, and Utku Evci. The dormant neuron phenomenon in deep reinforcement learning. In International Conference on Machine Learning (ICML), 2023.
>
> Rafael Rafailov, Tianhe Yu, Aravind Rajeswaran, and Chelsea Finn. Offline reinforcement learning from images with latent space models. In Learning for dynamics and control, 2021.
>
> Xinyue Chen, Che Wang, Zijian Zhou, and Keith W. Ross. Randomized Ensembled Double Q-Learning: Learning Fast Without a Model. In International Conference on Learning Representations (ICLR), 2021.
>
> Takuya Hiraoka, Takahisa Imagawa, Taisei Hashimoto, Takashi Onishi, and Yoshimasa Tsuruoka. Dropout Q-Functions for Doubly Efficient Reinforcement Learning. In International Conference on Learning Representations (ICLR), 2022.
>
> Wenhao Zhan, Baihe Huang, Audrey Huang, Nan Jiang, and Jason Lee. Offline reinforcement learning with realizability and single-policy concentrability. In Conference on Learning Theory (COLT), 2022.
>
> Volodymyr Tkachuk, Gellért Weisz, and Csaba Szepesvári. Trajectory Data Suffices for Statistically Efficient Learning in Offline RL with Linear $ q^\pi $-Realizability and Concentrability. Advances in Neural Information Processing Systems (NeurIPS), 2024.
>
> Abhishek Padalkar, Acorn Pooley, Ajinkya Jain, Alex Bewley, Alex Herzog, Alex Irpan, Alexander Khazatsky, Anant Rai, Anikait Singh, Anthony Brohan, et al. Open x-embodiment: Robotic learning datasets and rt-x models. arXiv preprint arXiv:2310.08864, 2023.
>
> Arsh Tangri, Ondrej Biza, Dian Wang, David Klee, Owen Howell, and Robert Platt. Equivariant offline reinforcement learning. arXiv preprint arXiv:2406.13961, 2024.
>
> Aviral Kumar, Rishabh Agarwal, Tengyu Ma, Aaron Courville, George Tucker, and Sergey Levine. Dr3: Value-based deep reinforcement learning requires explicit regularization. In The Tenth International Conference on Learning Representations (ICLR), 2022.
>
> Yang Yue, Rui Lu, Bingyi Kang, Shiji Song, and Gao Huang. Understanding, predicting and better resolving q-value divergence in offline-rl. In Advances in Neural Information Processing Systems (NeurIPS), volume 36, pp. 60247–60277, 2023.

---

> ### Comment · Reviewer_cV3D · 2024-11-25
>
> Thank you very much for addressing my (and other reviewers') comments and providing the clarifications. I find the updated manuscript to be much clearer overall, with well-presented and improved figures.

---

### Official Review · Reviewer_TcT9 · 2024-11-04

**Soundness:** 2
**Presentation:** 4
**Contribution:** 2
**Rating:** 3
**Confidence:** 5

**Summary:**

This paper introduces a novel offline-to-online reinforcement learning (O2O RL) approach tailored for robotic grasping with limited image-based demonstrations. The authors tackle the challenge of scarce data by developing a method, called Simplified Q, which integrates a regularization technique inspired by neural tangent kernel (NTK) theory, aiming to stabilize Q-value estimates and reduce the risk of Q-divergence. This method eliminates the need for a target network, which is traditionally used in actor-critic frameworks, allowing the agent to learn effectively with only 50 demonstrations and minimal online interaction. The approach is tested on a robotic grasping task with a vacuum-based manipulator and achieves a high success rate (>90%) in less than two hours, significantly outperforming behavioral cloning (BC) and other standard RL baselines. The paper demonstrates the algorithm’s robustness through ablation studies and comparative experiments, highlighting its potential for real-world applications where exploration is costly and data is limited​.

**Strengths:**

The proposed technique demonstrates impressive sample efficiency, allowing reinforcement learning to succeed with a limited number of demonstrations.

The paper provides strong empirical support through real-world experiments on image-based robotic manipulation, highlighting the practical applicability of the approach.

**Weaknesses:**

This work appears to reframe concepts that are well-explored in the reinforcement learning from demonstrations (RLfD) literature. The setup of offline-to-online reinforcement learning (O2O RL), which involves pretraining a policy with approximately 50 demonstrations and then continuing online RL (with optional further use of the demonstrations), as detailed on page 4, closely mirrors standard practices in RLfD. Many RLfD approaches similarly involve pretraining on demonstrations followed by online refinement of the policy using both interaction data and demonstration experiences, in both continuous control domains, e.g. [1][3][4][5], and discrete domains, e.g. [2].

To strengthen the contribution, it would be beneficial for the authors to provide a theoretical discussion on the relationship between O2O RL and RLfD. Additionally, incorporating experimental comparisons with established RLfD approaches would offer a more robust evaluation and ensure that the proposed method’s performance is rigorously benchmarked against strong baselines.

References:

[1] Vecerik, Mel, et al. "Leveraging demonstrations for deep reinforcement learning on robotics problems with sparse rewards." arXiv preprint arXiv:1707.08817 (2017).

[2] Hester, Todd, et al. "Deep q-learning from demonstrations." Proceedings of the AAAI conference on artificial intelligence. Vol. 32. No. 1. 2018.

[3] Zha, Y., Guan, L., & Kambhampati, S. (2024, March). Learning from ambiguous demonstrations with self-explanation guided reinforcement learning. In Proceedings of the AAAI Conference on Artificial Intelligence (Vol. 38, No. 9, pp. 10395-10403).

[4] Nair, Ashvin, et al. "Overcoming exploration in reinforcement learning with demonstrations." 2018 IEEE international conference on robotics and automation (ICRA). IEEE, 2018.

[5] Rajeswaran, Aravind, et al. "Learning complex dexterous manipulation with deep reinforcement learning and demonstrations." arXiv preprint arXiv:1709.10087 (2017).

**Questions:**

Do you have insights on the number of demonstration trajectories required for effective training?

---

> ### Author Response · Authors · 2024-11-18
>
> We thank reviewer TcT9 for the thoughtful feedback and we are delighted to see that reviewer TcT9 is impressed by our experimental results and the applicability of our method. Below we address their comments and questions and we hope they can reconsider our score:
>
> > discussion on the relationship between O2O RL and RLfD
>
> We thank reviewer TcT9 for noting the connection between RLfD and O2ORL—indeed RLfD and O2ORL are closely related. We have updated our related work section to include RLfD and also the provided references.
> Regarding the specifics of the references [1, 2, 3, 4, 5], we thank the reviewer for pointing these papers out. We note that besides [2] and [3], the other references operated in state-based observations. Our finding is that many existing algorithms conducted in state-based environments failed to transfer to image-based environments, as pointed out by Lu et al. (2023) and Rafailov et al. (2024), as well as a simple reaching task in Appendix B. Regarding [3], we thank the reviewer for providing this new result. We believe [3] is an orthogonal direction—if we understand correctly, SERLfD can also suffer from Q-value divergence provided that the underlying RL agent is Q-learning based. We further note that our approach includes minimal domain knowledge (i.e. no predicates) which improves the applicability of our approach. Nevertheless, we believe combining both SERLfD and Simplified Q for more complex tasks might be required.
>
> > experimental comparisons with established RLfD approaches
>
> We totally agree with reviewer TcT9 that comparing against RLfD baselines is important in demonstrating the robustness of our algorithms. We have attempted a similar approach to DQfD [2], replacing the DQN algorithm with Simplified-Q algorithm. The result can be found in Table 2a, w/o symmetric sampling (SS). We have noticed that simply including demonstrations in the offline buffer performs worse than using SS in both success rate and P-stop rate, corroborating the findings in RLPD (Ball et al., 2023). Therefore we have decided that RLPD is a more competitive baseline over DQfD (or SACfD in this case). In particular we have compared against RLPD built upon SAC, CrossQ, and two new additional algorithms SAC + LayerNorm (Yue et al., 2023) and SAC + DR3 (Kumar et al., 2022), all of which include a CQL regularizer due to the extra offline RL step. The results can be found in Figures 2 and 3—we have run each algorithm across three seeds and obtained the bootstrapped confidence interval—we can see that SimplifiedQ is better than all compared algorithms.
>
> Finally, while we want to conduct more experiments to further strengthen our claim, we emphasize that these are real-life robotic experiments—due to the cost of running the experiments in both time and robot damages, comparing against older baselines before RLPD might be unsuitable. As an alternative we have conducted offline RL experiments in robomimic (Table 1), as suggested by reviewers cV3D and awMg. Based on the results we can observe that Simplified Q is better than CQL on majority of the tasks, but similar to our original findings none of the offline RL algorithms is competitive against BC.
>
> > insights on the number of demonstration trajectories required for effective training
>
> We thank reviewer TcT9 for this question. For BC we have surprisingly found that it needs more than 500 demonstrations (10x) with data augmentation in order to achieve above 60% grasp success rate (Figure 2, right). Regarding offline RL, we have found that having at least 50 is sufficient for the policy to start approaching the bin, albeit missing the item of interest occasionally—we believe this is mostly due to the coverage of the space. Behaviourally, even with 50 demonstrations the offline RL agent is unable to go towards a specific corner of the bin initially, not until the online RL phase does the agent end up learning to grasp from said corner.

---

> > ### Comment · Reviewer_TcT9 · 2024-12-02
> > **Response to Authors**
> >
> > Thank you for your detailed rebuttal, additional experiments, and discussion of the relationship between your method and RLfD. I have carefully reviewed your response and the revised manuscript. While I appreciate the effort put into addressing my comments, I still find the overall positioning of the paper unclear, particularly regarding the distinctions and connections between offline-to-online RL, RLfD, and visual RL. My concerns are outlined below:
> >
> > **1. Confusion Between Offline RL and RLfD:**
> >
> > The motivation and evaluation settings of this paper blur the line between offline RL and RLfD. In offline-to-online RL, the offline data typically consists of suboptimal or diverse-quality experiences, which necessitate offline RL techniques rather than RLfD or behavioral cloning. However, your evaluation setting—using 50 successful human-teleoperated demonstrations for pretraining—aligns more closely with RLfD evaluation than traditional offline RL. Additionally, while the related work section states that your approach is "agnostic to the quality of data" and does not use behavioral cloning objectives, no experimental evidence supports the advantages of excluding behavioral cloning losses.
> >
> > To address this, I recommend two potential directions:
> >
> > A) Distinguish this work from RLfD: Focus on demonstrating the method’s effectiveness with lower-quality or mixed-quality data (as is typical in offline RL), and modify the evaluation to include both demonstration and unsuccessful offline data.
> >
> > B) Reposition the paper as RLfD research: Revise the motivation, introduction, and related work to frame the method as an advancement in RLfD. This would align the narrative with the evaluation setup and better situate the contributions within the RLfD field.
> >
> > **2. Evaluation Concerns for RLfD Comparisons:**
> >
> > If the paper is motivated as RLfD research, the evaluation should more rigorously compare the proposed method against state-of-the-art RLfD techniques. While the additional experiments on DQfD are appreciated, and RLPD includes SACfD as a baseline, these do not represent cutting-edge RLfD methods. For example, earlier works like [3] include more sophisticated techniques, such as environment resetting from demonstrations and additional behavior cloning losses, which are missing in your comparisons. There are also more recent RLfD works, such as [4-8], which should be re-implemented and adapted to image-based tasks (e.g., using convolutional nets or diffusion policies). These would provide a more robust and convincing benchmark for your method.
> >
> > **3. Motivation and Connection to Visual RL:**
> >
> > The connection between the proposed Q-regularization method and visual RL is unclear. The motivation to "stabilize visual-based RL algorithms to enable sample-efficient RL on real-life robotics tasks" does not align with the core technique. From a theoretical perspective, the Q-regularizer is agnostic to the nature of the input (pose or visual states) and is not inherently tied to enhancing visual processing. In contrast, works focused on advancing visual RL (e.g., diffusion policies [1], visual transformers for RL [2]) directly tackle visual processing challenges in RL.
> >
> > If the visual benefits of the proposed technique are to be claimed, the evaluation should include comparisons with state-of-the-art visual RL methods, particularly those with advanced visual representations or policies (e.g., [1][2]).
> >
> > **4. Ablation Studies on Q-Regularization:**
> >
> > The paper claims significant benefits from replacing the target network with the proposed Q-regularizer. However, these benefits are not adequately demonstrated. An ablation study comparing Simplified Q with and without the regularizer would clarify its impact. Additionally, comparing Simplified Q against methods using target networks in similar conditions would strengthen the claim that the proposed regularizer simplifies and improves Q-learning.

---

> > > ### Comment · Reviewer_TcT9 · 2024-12-02
> > > **Cont’d Response to Authors**
> > >
> > > **References**
> > >
> > > [1] Kang, Bingyi, et al. "Efficient diffusion policies for offline reinforcement learning." Advances in Neural Information Processing Systems 36 (2024).
> > >
> > > [2] Hansen N, Su H, Wang X. Stabilizing deep q-learning with convnets and vision transformers under data augmentation[J]. Advances in neural information processing systems, 2021, 34: 3680-3693.
> > >
> > > [3] Nair, Ashvin, et al. "Overcoming exploration in reinforcement learning with demonstrations." 2018 IEEE international conference on robotics and automation (ICRA). IEEE, 2018.
> > > [4] Hansen, Nicklas, et al. "Modem: Accelerating visual model-based reinforcement learning with demonstrations." arXiv preprint arXiv:2212.05698 (2022).
> > >
> > > [5] Huang Z, Wu J, Lv C. Efficient deep reinforcement learning with imitative expert priors for autonomous driving[J]. IEEE Transactions on Neural Networks and Learning Systems, 2022, 34(10): 7391-7403.
> > >
> > > [6] Zha Y, Guan L, Kambhampati S. Learning from ambiguous demonstrations with self-explanation guided reinforcement learning[C]//Proceedings of the AAAI Conference on Artificial Intelligence. 2024, 38(9): 10395-10403.
> > >
> > > [7] Ramirez, Jorge, and Wen Yu. "Reinforcement learning from expert demonstrations with application to redundant robot control." Engineering Applications of Artificial Intelligence 119 (2023): 105753.
> > >
> > > [8] Coelho, Daniel, Miguel Oliveira, and Vitor Santos. "RLfOLD: Reinforcement Learning from Online Demonstrations in Urban Autonomous Driving." Proceedings of the AAAI Conference on Artificial Intelligence. Vol. 38. No. 10. 2024.

---

> ### Author Response · Authors · 2024-11-18
>
> References:
> Philip J Ball, Laura Smith, Ilya Kostrikov, and Sergey Levine. Efficient online reinforcement learning with offline data. In International Conference on Machine Learning (ICML), volume 202, pp. 1577–1594, 2023.
>
> Aviral Kumar, Rishabh Agarwal, Tengyu Ma, Aaron Courville, George Tucker, and Sergey Levine. Dr3: Value-based deep reinforcement learning requires explicit regularization. In The Tenth International Conference on Learning Representations (ICLR), 2022.
>
> Cong Lu, Philip J. Ball, Tim G. J. Rudner, Jack Parker-Holder, Michael A Osborne, and Yee Whye Teh. Challenges and opportunities in offline reinforcement learning from visual observations. Transactions on Machine Learning Research, 2023.
>
> Rafael Rafailov, Kyle Beltran Hatch, Anikait Singh, Aviral Kumar, Laura Smith, Ilya Kostrikov, Philippe Hansen-Estruch, Victor Kolev, Philip J. Ball, Jiajun Wu, Sergey Levine, and Chelsea Finn. D5RL: Diverse datasets for data-driven deep reinforcement learning. Reinforcement Learning Journal, 5:2178–2197, 2024.
>
> Yang Yue, Rui Lu, Bingyi Kang, Shiji Song, and Gao Huang. Understanding, predicting and better resolving q-value divergence in offline-rl. In Advances in Neural Information Processing Systems (NeurIPS), volume 36, pp. 60247–60277, 2023.

---

> > ### Author Response · Authors · 2024-11-27
> > **Kind reminder**
> >
> > Dear Reviewer TcT9,
> >
> > We thank you for the time spent reviewing our submission.
> >
> > As the main discussion phase is ending soon, we wanted to send this gentle reminder. We have done our best to answer the comments you raised, as well as incorporate your suggestions. We would love to hear back from the reviewer and whether we have addressed their concerns.

---

> ### Author Response · Authors · 2024-12-02
> **Response to further concerns**
>
> We thank reviewer TcT9 for their detailed response, we greatly appreciate the discussions on O2ORL and RLfD.
>
> > C1
>
> Thank you for the discussion. We would like to express that O2O RL and RLfD are very similar except for the fact that RLfD usually includes near-optimal data while potentially removing the assumption of having reward access, as mentioned in the previous response. In robotic settings, demonstrations are often provided as suboptimal data (e.g. RoboMimic and our real-life experiments), not random motion like Brownian motion, as that is harmful to the robot. Secondly, we do not claim all demonstrators to be equally good—some demonstrators can result in significantly lower performance in practice—while all trajectories were successes, we cannot claim that the sole demonstrator is optimal.
> Nevertheless, due to very limited time constraints, we have only conducted experiments with mixed data on D4RL’s relocate and door environments. According to D4RL’s results, CQL and Simplified Q will perform slightly better than BC due to the mixture of data, furthermore Simplified Q does better than CQL. We again emphasize Simplified Q being better has been consistent across all benchmarks that we were able to test on and we truly do not believe this can be due to luck.
>
> |               	| CQL                 	| Simplified Q      	|
> |-------------------|-------------------------|-----------------------|
> | `relocate-cloned` | 0.043  +/-  0.003   | 0.061  +/-  0.008 |
> | `door-cloned` 	| -0.209  +/-  0.029 |  0.065  +/-  0.012  |
>
> > C2 and C3
>
> Thank you for the extra references. We sincerely appreciate the reviewer’s effort on providing these results. While we are not motivated by RLfD, we would like to raise that *none of these works* were evaluated in real-life environments. In real-life there are considerations including inference time—in practice many of existing papers that leverage larger models including diffusion policies/transformers inference at <10Hz frequency (e.g. diffusion policy, RT-X). In fact we did try OpenVLA as well and it showed subpar performance and not any better than our BC baseline with a standard ResNet model.
>
> Finally, the surprising result and its significance is that although D5RL has demonstrated existing RL methods designed on state-based environments fail on image-based environments, specifically robotic manipulation, our method can still succeed without any image-specific nor robotic-specific techniques (we note that D5RL has not yet open-sourced their dataset, hence we were unable to directly compare). While the line of work using data augmentation or pretraining with large image-dataset are promising, our result suggests that these domain-specific techniques are unnecessary. This further enabled us to run real-life image-based robotic manipulation with RL in less than few hours, which is not common at all except for limited work such as CQN (QT-Opt and SAC-X they have significantly larger scale experiments). We emphasize once again that running on real-life environments is significantly more challenging than in simulation due to cost and scalability.
>
> > C4
>
> Please see appendix C.2 for ablation on whether our regularizer makes an impact and other design decisions including $N$-step return and symmetric sampling. Secondly, all of our baselines leverage the same algorithmic techniques such as symmetric sampling and 3-step returns as mentioned on lines 255-257. We note that all of these experiments are conducted on real-life environments.

---

### Author Response · Authors · 2024-11-18
**General Response to the Reviewers**

We thank all reviewers (TcT9, cV3D, L1Gj, and awMg) for their feedback. We are pleased to see that the results from the real-life robotic task are highly appreciated and interesting (reviewers TcT9, cV3D, awMg). We are also delighted to see that our analyses are well motivated and sound (reviewers TcT9, cV3D, awMg).

Here are a few common comments regarding this work that we will address altogether. All the updates in the manuscript are reflected in blue text.
1. We understand that all reviewers are concerned about the effectiveness of Simplified Q due to the single real-life robotic experiment. Therefore we have included a few additional results. We believe the reviewers will be receptive to the extra results and find our work more convincing than previously, and as a result reconsider their scores accordingly.
    - We have included two more RL algorithms that aim to stabilize Q-divergence, namely DR3 (Kumar et al., 2022) and SAC with LayerNorm (Yue et al., 2023). Furthermore, we have compared all algorithms across three seeds rather than a single seed, with the goal of providing more convincing statistical results. All results are presented in Figures 2 and 3 of the updated manuscript. Generally, the results remain consistent with our original claim where Simplified Q can outperform the compared algorithms with a similar number of total data. Even with the similarity with DR3, we have shown that Simplified Q is more sample efficient than DR3 and further experiences less P-stops over the training run.
    - We have compared BC, CQL, and Simplified Q on a simulated environment with low-dimensional observation, robomimic (Mandlekar et al., 2022). Specifically the algorithms are evaluated on lift, can, and square, with both proficient-human (PH) and multi-human datasets. We again compare all algorithms across three seeds. We directly add the regularizer on the CQL implementation and remove the target network—Simplified Q uses the same hyperparameters as the original CQL implementation for this experiment. The implementation can be found here: https://anonymous.4open.science/r/robomimic-3DB6/robomimic/. The results are presented in Table 1 of the updated manuscript. Similar to our findings in Figure 1, middle, offline RL algorithms are inferior to BC. However, we can still observe that Simplified Q can generally outperform, if not comparable to CQL, again corroborating our findings in the real-life experiment.
    - Finally, we conducted ablation studies and sensitivity analyses on the newly introduced $\beta$ in appendix C.2. We hope all reviewers can take into consideration the effort required for running a real-life robotic experiment—it is difficult to scale due to hardware limitations and require significant human time in case of unexpected situations, e.g., the robotic arm is unable to recover from a pose due to collision.
1. Reviewers TcT9, cV3D, and L1Gj have raised questions about the differences and benefits between Simplified Q and previous work (e.g. DR3 and methods from RLfD).
    - Generally, we believe O2ORL and RLfD are very similar—O2ORL usually imposes no assumptions on the offline dataset (aside from having reward signal) while RLfD usually assumes the offline dataset is near optimal (but may exclude reward signal). This means that in O2ORL using BC as part of the policy optimization objective is undesirable and may require extra care. Unlike methods like TD3-BC (Fujimoto and Gu, 2021) and CQN (Seo et al., 2024), Simplified Q does not leverage any form of BC in the learning objectives. We also note that RLPD (Ball et al., 2023) have demonstrated that SACfD is inferior to RLPD for a wider range of tasks.
    - Our work heavily leverages similar analyses done in previous studies, as pointed out in sections 3 and 5. However, our insight is that this regularizer can simplify Q-learning by removing the target network which reduces half of the Q-parameters and reduces the number of related hyperparameters into a single scalar. On the other hand, other works complicate their algorithms by introducing extra components and hyperparameters which make deployment to real-life systems difficult—one will have to tune more hyperparameters which is often infeasible in real-life applications. We also found that generally our hyperparameter is quite robust to a range of values between [0.1, 0.4] as shown in Table 2c.
    - As far as we know, we are one of the first to successfully apply O2ORL in real-life image-based robotic manipulation without any behavioural cloning nor hand-designed heuristics (e.g. predicates or hierarchies). This is promising as this is an important step towards enabling self-improvement on real-life robotic systems.

---

> ### Author Response · Authors · 2024-11-18
> **Result on Robomimic**
>
> The result on robomimic is also presented here (Table 1 in updated manuscript). Here we can see that Simplified Q is generally better than CQL except for lift, PH, where we observe a divergence on one seed. Nevertheless, similar to the general message of the original paper, we have found that offline RL is inferior to BC with the only given demonstrations, and Simplified Q is better than CQL.
>
> |             | BC                         | CQL              | Simplified Q (Ours) | Ours w/ Wider Critic |
> |-------------|----------------------------|------------------|---------------------|---------------------|
> | Lift (MH)   | **100.00 +/- 0.00** | 82.00 +/- 6.18 | 98.00 +/- 1.63    | 99.33 +/- 0.54 |
> | Can (MH)    | **84.00 +/- 2.49**  | 26.67 +/- 5.68 | 35.33 +/- 14.62   | 37.33 +/- 4.65 |
> | Square (MH) | **47.33 +/- 0.54**  | 1.33 +/- 1.09  | 5.33 +/- 2.18     | 12.00 +/- 2.49 |
> | Lift (PH)   | **100.00 +/- 0.00** | 95.33 +/- 3.81 | 78.67 +/- 16.61   | 100.00 +/- 0.00 |
> | Can (PH)    | **94.67 +/- 1.09**  | 37.33 +/- 4.84 | 87.33 +/- 3.57    | 91.33 +/- 2.88 |
> | Square (PH) | **82.00 +/- 0.94**  | 4.67 +/- 0.54  | 7.33 +/- 3.03     | 6.67 +/- 5.44 |

---

> ### Author Response · Authors · 2024-12-02
> **Result on Adroit**
>
> As the discussion period is coming to an end, we would like to thank all reviewers for their discussions and suggestions.
>
> We have received extra feedback regarding using mixture data to demonstrate further robustness (reviewer TcT9), and have conducted experiments on D4RL's `relocate-cloned` and `door-cloned` environments which are 50% human data and 50% policy data. We note that the human data is significantly worse than the policy data in this case. We ran CQL and Simplified Q on three seeds and have observed that Simplified Q is still consistent better (please see table below).
>
> |               	| CQL                 	| Simplified Q      	|
> |-------------------|-------------------------|-----------------------|
> | `relocate-cloned` | 0.043  +/-  0.003   | 0.061  +/-  0.008 |
> | `door-cloned` 	| -0.209  +/-  0.029 |  0.065  +/-  0.012  |

---

### Meta-Review · Area_Chair_nqji · 2024-12-18

**Metareview:**

This paper presents an offline-to-online reinforcement learning approach for image-based robotic grasping, introducing a simplified Q-learning method that replaces the target network with an NTK-inspired regularization technique. Despite some interesting elements, the reviewers identified several critical limitations.

The primary concern, raised by multiple reviewers, was the paper's unclear positioning and limited technical novelty. While the authors demonstrated good results on a robotic grasping task, Reviewer awMg noted that the proposed NTK regularization appeared to be a general RL approach rather than one specifically designed for robotics, yet lacked validation on standard RL benchmarks like D4RL. This fundamental disconnect between the method's generality and the paper's robotic focus remained unresolved in the rebuttal.

Reviewer L1Gj's detailed analysis identified the main technical contributions as relatively minor modifications to existing approaches - namely, pushing towards zero correlation rather than negative correlation (compared to DR3), and simplifying OOD policy handling. These incremental changes, while potentially useful, do not represent a significant advancement in the field.

While the authors made efforts during the rebuttal to address clarity issues raised by Reviewer cV3D and added additional baselines, Reviewer TcT9 highlighted a more fundamental problem with the paper's relationship to reinforcement learning from demonstrations (RLfD). The added baseline comparisons (DQNfD, SACfD) were not state-of-the-art, and the paper failed to properly position itself within the broader context of RLfD research.

This AC has carefully considered the authors' response to TcT9's review but find the criticism valid - the paper's positioning remains unclear and the technical contributions, while potentially useful, are too incremental. While the real-world results are promising, the lack of broader validation, limited technical novelty, and unclear relationship to existing literature make this work more suitable for refinement and resubmission.

**Additional Comments On Reviewer Discussion:**

None (included in metareview)

---

### Decision · Program_Chairs · 2025-01-22

Reject